# Cortactin stabilizes actin branches by bridging activated Arp2/3 to its nucleated actin filament

**Tianyang Liu** [1], **Luyan Cao** [2], **Miroslav Mladenov**[2], **Antoine Jegou** [3], **Michael Way** [2,4] ✉ & **Carolyn A. Moores** [1] ✉

Regulation of the assembly and turnover of branched actin filament networks nucleated by the Arp2/3 complex is essential during many cellular processes, including cell migration and membrane trafficking. Cortactin is important for actin branch stabilization, but the mechanism by which this occurs is unclear. Given this, we determined the structure of vertebrate cortactin-stabilized Arp2/3 actin branches using cryogenic electron microscopy. We find that cortactin interacts with the new daughter filament nucleated by the Arp2/3 complex at the branch site, rather than the initial mother actin filament. Cortactin preferentially binds activated Arp3. It also stabilizes the F-actin-like interface of activated Arp3 with the first actin subunit of the new filament, and its central repeats extend along successive daughter-filament subunits. The preference of cortactin for activated Arp3 explains its retention at the actin branch and accounts for its synergy with other nucleation-promoting factors in regulating branched actin network dynamics.

The actin cytoskeleton can form many types of dynamic supramolecular array—from linear bundles to branched actin filament networks, which underlie its functional diversity and adaptability[1–6]. Distinct F-actin arrays are formed by the localized activities of specific actin nucleating factors, actin-binding proteins and myosin motors[1,2,7].

Branched actin networks are generated when a new 'daughter' filament is nucleated from the side of a pre-existing 'mother' filament by the seven-subunit Arp2/3 complex that contains actin-related protein 2 (Arp2) and Arp3 (refs. 2,8,9). Activation of the Arp2/3 complex involves conformational rearrangement of the complex, resulting in the formation of a short-pitch helical F-actin-like template. The fast-growing barbed end of the daughter filament extends from this template[10,11]. Class 1 nucleation-promoting factors (NPFs), such as WAVE and WASP, activate Arp2/3 through their conserved carboxy-terminal VCA domain, consisting of one to three verprolin domains (also known as WASP-homology 2 domains) followed by central and acidic

segments[12–20]. The VCA domains of class 1 NPFs also stimulate nucleation by recruiting actin subunits to the activated Arp2/3 complex, from which these NPFs are subsequently released (Extended Data Fig. 1a)[21–23].

The correct functioning of Arp2/3-nucleated branched actin networks depends on not only their spatial and temporal assembly but also their stability and turnover[24–28]. The actin-binding protein cortactin, which is considered a class 2 NPF, plays a crucial part in stabilizing actin branches. It interacts with the Arp2/3 complex and actin filaments through its amino-terminal acidic domain (NtA) and 6.5 central unstructured 37-amino-acid repeats, respectively[29], although the molecular basis of these interactions is unclear (Fig. 1a)[30–33]. Furthermore, although cortactin alone can weakly activate the Arp2/3 complex, it synergizes with class 1 NPFs to further stimulate efficient Arp2/3-mediated formation of actin branches[21,22,34]. Given its central role in stabilizing branched actin networks, cortactin is important in many cellular processes such as epithelial integrity and intracellular

[1]Institute of Structural and Molecular Biology, Birkbeck College, London, UK. [2]The Francis Crick Institute, London, UK. [3]Université Paris Cité, CNRS, Institut Jacques Monod, Paris, France. [4]Department of Infectious Disease, Imperial College, London, UK. ✉e-mail: Michael.Way@crick.ac.uk; c.moores@bbk.ac.uk

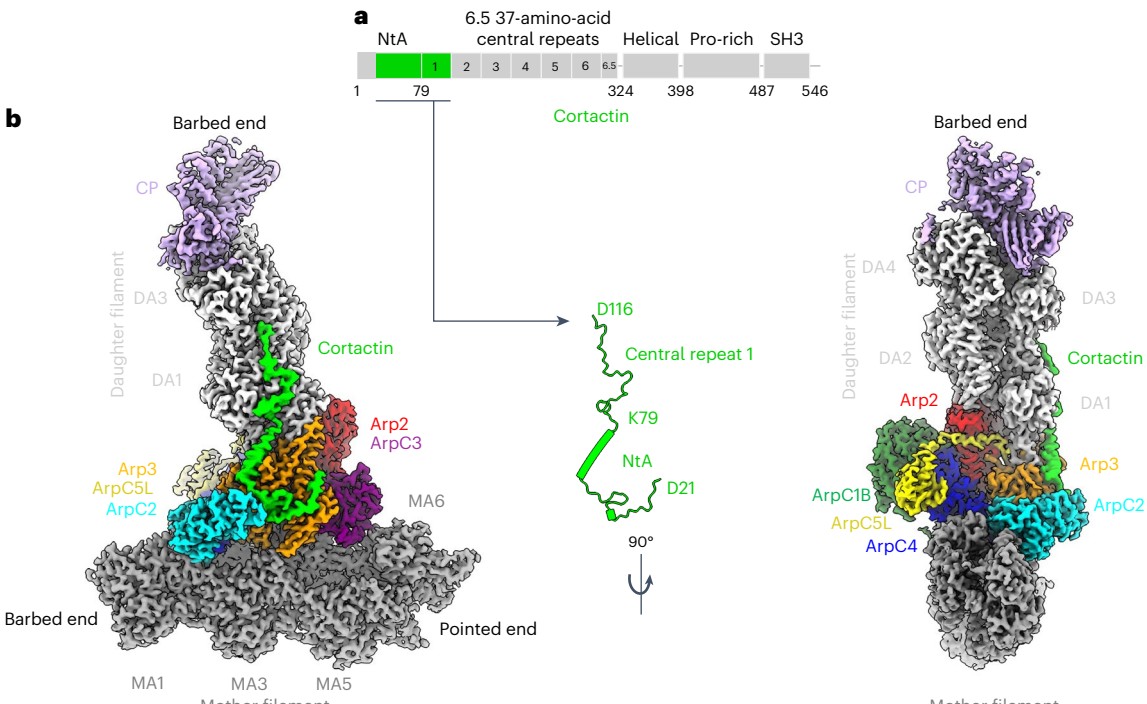

**Fig. 1 | Cortactin binds the daughter filament at Arp2/3-mediated actin branches. a**, Cortactin domain organization. **b**, Overview of the composite cryo-EM reconstruction of the cortactin-stabilized Arp2/3 actin branch, assembled from four local refined reconstructions, as shown in Extended Data Figs. 2 and 3. Densities of individual proteins are colored according to the labels, and the mother- and daughter-filament subunits are colored dark and light gray, respectively, and are labeled MA1, MA3, MA5 and MA6 and DA1–DA4. The free barbed and pointed ends of the mother and daughter filaments are also labeled. The central inset shows the cortactin model calculated from the cryo-EM reconstruction, with the visualized regions of the full-length protein mapped on to the cortactin schematic in **a**, as indicated.

trafficking, as well as a range of pathologies, including bacterial infection and cancer metastasis[29,35–37]. However, despite its functional importance, the precise mode of action of cortactin and its mechanism of synergy with class 1 NPFs remain unknown.

## Results

### Cortactin connects the Arp2/3 complex to its nucleated daughter filament

To maximize the number of branches in our sample, therefore increasing the possibility of visualizing the hitherto elusive binding site of cortactin, we used the most active isoform of human Arp2/3 (Arp2/3-C1B-C5L[33]) and included capping protein in our sample to limit daughter-filament growth[38–41]. The resulting cryogenic electron microscopy (cryo-EM) structure of cortactin-stabilized Arp2/3 actin branches had an overall resolution of approximately 3.3 Å, allowing us to visualize cortactin (Fig. 1b, Table 1, Extended Data Figs. 1–3 and Supplementary Video 1). This structure showed that, unexpectedly, cortactin connects the activated Arp2/3 complex and the daughter filament, in contrast to previous proposals suggesting that cortactin binds to the mother filament (Fig. 1b)[22,34]. In the presence of cortactin, the overall conformation of the activated Arp2/3 complex at the junction of mother and daughter filaments is similar to that of previous cryo-EM structures[10,42,43] (Extended Data Fig. 4a,b and Supplementary Video 1). The daughter filament consists of four subunits (DA1–DA4), each bound to ADP, and its barbed end is terminated by capping protein[38–41]. The cortactin density that extends along this short daughter filament corresponds to the first cortactin repeat (Fig. 1b). No density corresponding to cortactin is observed on the ADP-bound mother filament (consisting of MA1–MA6 in our image-processing scheme; Extended Data Fig. 2). Both mother and daughter filaments adopt canonical ADP-F-actin structures[44,45], and Arp2 and Arp3 are also bound to ADP (Extended Data Fig. 4b–d). Overall, our structure shows that, rather than modifying the filaments at actin branches, cortactin branch stabilization is mediated by the protein–protein contacts that cortactin forms with the activated Arp2/3 complex and the daughter filament[30–32].

### Cortactin preferentially binds to activated Arp3

Cortactin NtA domain residues 21–79 form electrostatic and hydrophobic interactions with all four Arp3 subdomains, as well as contacting ArpC2 (Fig. 2a and Extended Data Fig. 5a–d). The cortactin D21-W22-E23 motif, which is essential for the interaction with Arp2/3 (ref. 31), inserts into a positively charged pocket of Arp3, and residues 24–54 adopt a meandering trajectory across the Arp3 surface (Fig. 2a,b and Extended Data Fig. 5). At residue 55, the NtA domain turns by approximately 90° on the surface of Arp3 and forms an amphipathic α-helix (residues I55–T76), which binds in a hydrophobic cleft on Arp3 and points towards the daughter filament (Fig. 2a,c). This cortactin helix binds adjacent to the Arp3 hinge helix (residues 145–154), which is key in mediating inactive–active Arp2/3 complex structural transitions[10,11,43].

The cortactin NtA α-helix stabilizes the Arp3 W-loop (residues 180–187) in a conformation that has been previously observed only in activated Arp2/3 and is distinct from that seen in the inactivated Arp2/3 complex[10,20,46,47]. Consequently, the structural groove at the barbed end of Arp3 is open, promoting interaction with DA1 of the new daughter filament (Fig. 2d). The contacts formed between activated Arp3 and DA1 mimic the longitudinal contacts along F-actin[10,11] and involve insertion of subdomain 2 of DA1—specifically its 'D-loop'—in the barbed end of Arp3 (Fig. 2d). Further, the loop within Arp3 (residues 155–164) that follows the hinge helix—which we now term the cortactin loop—makes contacts with the cortactin NtA α-helix through a distinct conformation compared with branch structures in the absence of cortactin (Fig. 2e). The structure is consistent with a model in which the interaction of the

**Table 1 | Cryo-EM data collection, 3D image processing and model building statistics**

| | Mother filament consensus map (EMDB-17558) | Daughter filament consensus map (EMDB-17553) | Locally refined map of mother filament (EMDB-17557) (PDB 8P94) | Locally refined map of Arp2/3 complex and cortactin (EMDB-17554) (PDB 8P94) | Locally refined map of daughter filament and cortactin (EMDB-17555) (PDB 8P94) | Locally refined map of capping protein and DA4 (EMDB-17556) (PDB 8P94) |
|---|---|---|---|---|---|---|
| **Data collection and processing** | | | | | | |
| Magnification | ×81,000 | ×81,000 | ×81,000 | ×81,000 | ×81,000 | ×81,000 |
| Voltage (kV) | 300 | 300 | 300 | 300 | 300 | 300 |
| Electron exposure (e⁻/Å²) | 49.4 | 49.4 | 49.4 | 49.4 | 49.4 | 49.4 |
| Defocus range (μm) | −0.9 to −2.4 | −0.9 to −2.4 | −0.9 to −2.4 | −0.9 to −2.4 | −0.9 to −2.4 | −0.9 to −2.4 |
| Pixel size (Å) | 1.067 | 1.067 | 1.067 | 1.067 | 1.067 | 1.067 |
| Symmetry imposed | $C_1$ | $C_1$ | $C_1$ | $C_1$ | $C_1$ | $C_1$ |
| Initial particle images (no.) | 2,001,580 | 2,001,580 | 2,001,580 | 2,001,580 | 2,001,580 | 2,001,580 |
| Final particle images (no.) | 130,915 | 179,923 | 130,915 | 179,923 | 179,923 | 176,179 |
| Map resolution (Å) | 3.3 | 3.3 | 3.3 | 3.2 | 3.3 | 3.8 |
| FSC threshold | 0.143 | 0.143 | 0.143 | 0.143 | 0.143 | 0.143 |
| Map resolution range (Å) | | | 3.0–6.5 | 2.9–5.2 | 3.0–7.3 | 3.5–6.9 |
| **Refinement** | | | | | | |
| Initial model used | | | AlphaFold[62,63] | AlphaFold[62,63] | AlphaFold[62,63] | AlphaFold[62,63] |
| Model resolution (Å) | | | 3.7 | 3.7 | 3.8 | 4 |
| FSC threshold | | | 0.5 | 0.5 | 0.5 | 0.5 |
| Map sharpening B factor (Å²) | | | −75.6 | −78.4 | −81 | −112.3 |
| Model composition | | | | | | |
| Non-hydrogen atoms | | | 33,205 | 36,301 | 30,001 | 13,405 |
| Protein residues | | | 2,262 | 2,436 | 2,054 | 918 |
| Ligands | | | 24 | 6 | 16 | 2 |
| B factors (Å²) | | | | | | |
| Protein | | | 105.4 | 40.43 | 81.29 | 73.29 |
| Ligand | | | 35.11 | 31.71 | 31.1 | 31.71 |
| R.m.s. deviations | | | | | | |
| Bond lengths (Å) | | | 0.008 | 0.005 | 0.008 | 0.007 |
| Bond angles (°) | | | 1.686 | 1.116 | 1.597 | 1.31 |
| **Validation** | | | | | | |
| MolProbity score | | | 1.5 | 1.62 | 1.69 | 1.58 |
| Clashscore | | | 9.24 | 6.84 | 11.67 | 8.45 |
| Poor rotamers (%) | | | 0 | 0 | 0 | 0 |
| Ramachandran plot | | | | | | |
| Favored (%) | | | 97.07 | 96.36 | 97.48 | 97.36 |
| Allowed (%) | | | 2.03 | 3.64 | 2.52 | 2.64 |
| Disallowed (%) | | | 0 | 0 | 0 | 0 |

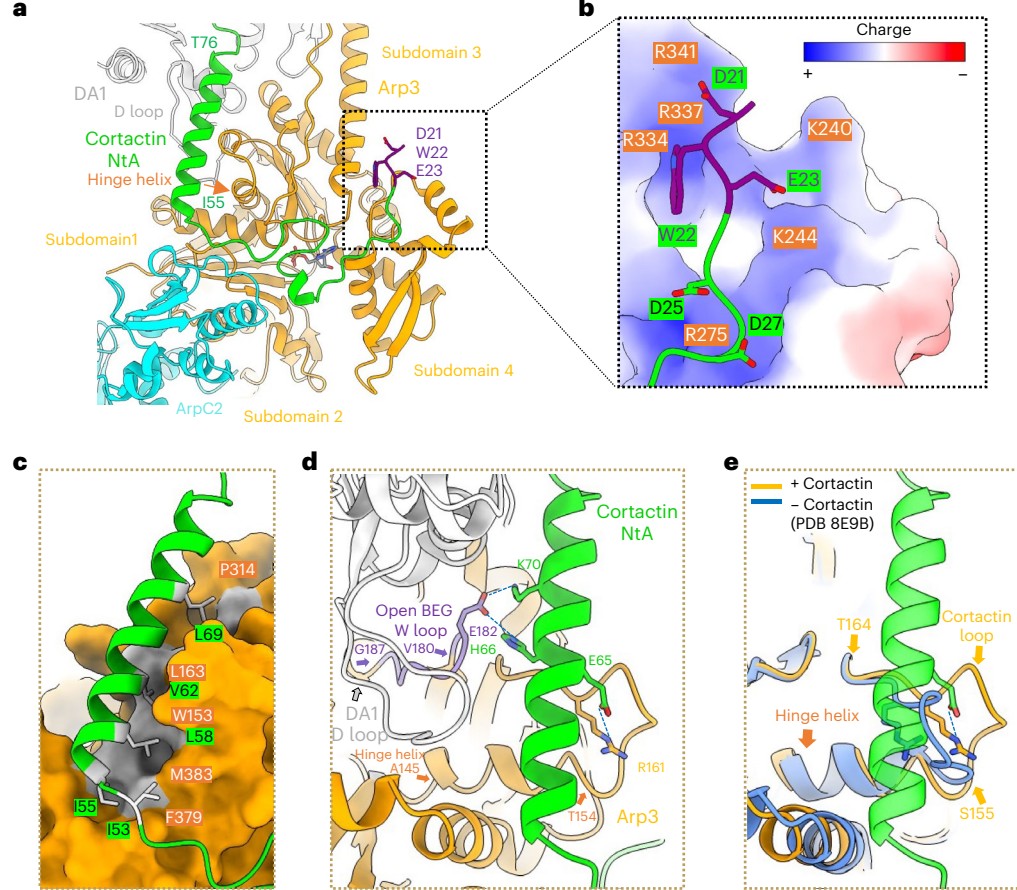

**Fig. 2 | Cortactin NtA binds to activated Arp3. a**, Overview of cortactin NtA (green and purple) and its interactions with Arp3 (orange) and ArpC2 (cyan). DWE motif (D21-W22-E23) residues are colored purple and shown as a stick model; the rest of cortactin NtA is depicted as a ribbon model. Cortactin residues D21–N54 meander across the Arp3 surface, and residues I55–T76 form an amphipathic α-helix. Subdomains in Arp3 are labeled, and the Arp3 hinge helix at the junction of subdomains 1 and 3 is indicated with an orange arrow. DA1 is the first subunit of the daughter filament, which, through the D-loop in its subdomain 2, forms longitudinal contacts with Arp3. Detailed views of interactions are shown in Extended Data Fig. 5a–d. **b**, Electrostatic interaction of the negatively charged cortactin N-terminal region that inserts into a positively charged pocket of Arp3. Arp3 is depicted in surface representation; acidic regions are shown in red, and basic regions are shown in blue with individual basic residues colored orange. Cortactin is depicted in a ribbon model, the DWE motif is shown as a purple stick

model and residues 24–29 are colored green with acidic residues shown as a stick model. **c**, The hydrophobic interaction of the cortactin NtA α-helix (green ribbon model), with a hydrophobic groove on the surface of Arp3 (orange space-filling representation, with hydrophobic regions in gray). Residues on the interaction surface of cortactin and Arp3 are labeled in green and orange, respectively. **d**, The cortactin NtA α-helix stabilizes the activated Arp3 W-loop (residues 180–187, purple) conformation to enable DA1 actin subunit binding through its D-loop in the open barbed-end groove (BEG) of Arp3. Residues forming salt bridges are shown in stick representation. The distances between interacting residues are provided in Extended Data Fig. 5a–d. **e**, The cortactin NtA α-helix (green) stabilizes and interacts with a specific conformation of the Arp3 loop (residues 155–164), which we term the cortactin loop (in orange). The loop adopts a distinct conformation in the presence of cortactin, and its conformation is different in activated Arp2/3 in the absence of cortactin (blue ribbon).

cortactin NtA domain with the activated Arp2/3 complex stabilizes the interface of Arp3 with DA1 of the daughter filament.

### Cortactin central repeats extend across daughter filament subunits

The first cortactin central repeat (residues 80–116) extends longitudinally by approximately 5.5 nm from the C terminus of the NtA α-helix on the daughter filament, bridging across the two successive subunits DA1 and DA3 (Fig. 3a). D116, the final residue of the first cortactin central repeat, is positioned on DA3 in a corresponding position to that of the first repeat residue (A80) on DA1 (Fig. 3a). Because of the short daughter filaments in our sample, we did not visualize more of the central repeats that were present in our full-length cortactin construct. However, on the basis of the cortactin F-actin interaction observed in our structure, we constructed a model illustrating how the cortactin central repeats would interact with a longer daughter filament. Our model shows that the second repeat would bind along the daughter filament in the same

way as the first repeat does, bridging across DA3 and DA5 (Fig. 3b). Furthermore, the model predicts that the cortactin repeats, including the C-terminal half repeat, would extend to the barbed end of DA13, a half-turn of the F-actin helix (Fig. 3c and Supplementary Video 2). The conservation of interacting residues within the cortactin repeats is also consistent with the repeating pattern of interactions with the hydrophobic and hydrophilic regions of the F-actin surface (Fig. 3d). Given the conserved amino acid distribution between all 6.5 repeats and the observed binding pattern of the first central repeat, it is likely that cortactin repeats act together to maximize branch stability. The binding of cortactin to actin is regulated by the acetylation of cortactin lysine residues, including K87 in repeat 1 (ref. 48). In our structure, K87 points towards a negatively charged patch on DA1, and its acetylation is predicted to perturb this interaction (Extended Data Fig. 6). Our model thus shows how the conserved pattern of lysine acetylation in cortactin central repeats (marked with an asterisk in Fig. 3d) would reduce binding activity between cortactin and F-actin and thereby impede cell

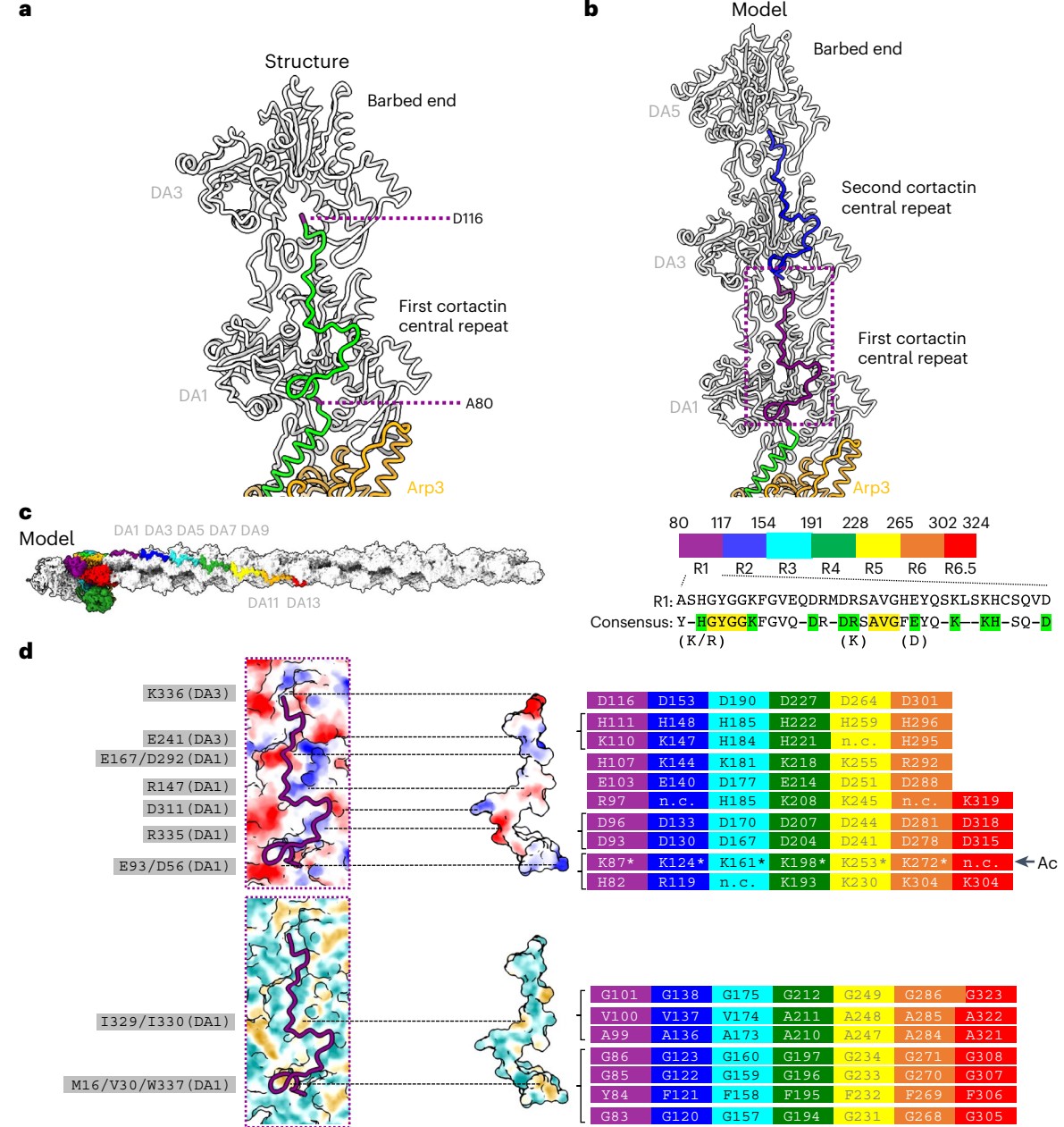

**Fig. 3 | Cortactin repeats bind the daughter filament. a**, The first cortactin repeat (in green) binds longitudinally along the daughter filament and forms a bridge between the DA1 and DA3 actin subunits (gray). The first and last residues of the first cortactin repeat are labeled and colored purple. See also Supplementary Video 2. **b**, Cortactin repeats are predicted to bind equivalent subunit positions along the same strand of the daughter filament. The cortactin NtA α-helix is shown in green; the first repeat, which forms a bridge between DA1 and DA3, is shown in purple; and the second repeat, which forms a bridge between DA3 and DA5, is shown in blue. **c**, Left, the 6.5 cortactin repeats are predicted to bind longitudinally along seven subunits of the daughter actin filament. The modeled cortactin central repeats are colored, from the N to C terminus, in purple, blue, cyan, green, yellow, orange and red. Right, amino acid sequence of the first repeat, and the consensus sequence of the 6.5 central repeats. Amino acid residues present in more than 5 of the 6.5 repeats are included in the consensus sequence. Charged residues (K, H and R and D and E) are grouped together in the analysis. Conserved residues in green form potential electrostatic interactions with actin subunits. Conserved

residues in yellow form potential hydrophobic interactions with actin subunits, as indicated in **d**. **d**, Top, the binding surface of DA1 actin and the cortactin first central repeat, colored by electrostatic potential and shown in open-book representation. Blue, positively charged; red, negatively charged. Conserved charged residues in cortactin are shown and colored according to the individual central repeat that they are in, as in **c**, with asterisks marking individual residues that are subject to acetylation, and the arrow and label Ac highlighting the conserved pattern of modification[48]; actin residues at the interface are listed on the left; residues within the same region are separated by '/'. Dotted lines indicate interaction regions in the assembly. Bottom, the binding surface of DA1 actin and the first central repeat of cortactin, colored by hydrophobicity and shown in open-book representation. Yellow, hydrophobic; cyan, hydrophilic. Conserved hydrophobic residues are shown and colored according to the individual central repeat that they are in, as in **c**. Actin residues at the interface are listed on the left; residues within the same region are separated by '/'; n.c., not conserved. Dotted lines indicate interaction regions in the assembly.

motility[48]. The observation that cortactin binds exclusively along the daughter filament also explains why cortactin stabilizes linear actin filaments nucleated by SPIN90–Arp2/3 complexes in the absence of a mother filament[49].

## Combined interaction of cortactin NtA and central repeat domain maximizes branch stabilization

Our data show that the daughter filament is stabilized directly by the cortactin repeats, which bind along the intra-strand subunits of the

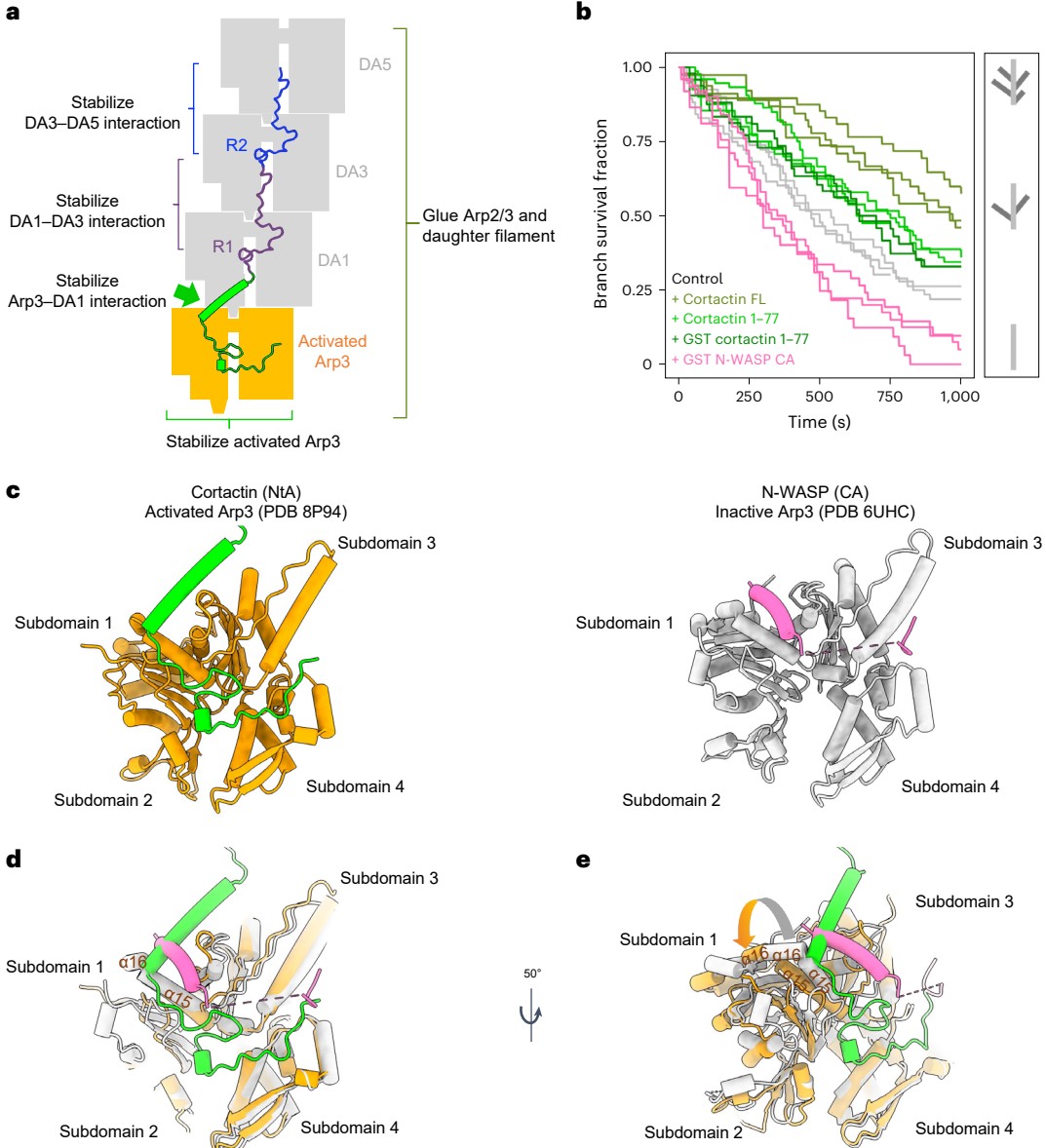

**Fig. 4 | The binding site of NtA cortactin on Arp3 explains its synergy with VCA domains. a**, Schematic showing how cortactin stabilizes the actin branch junction. **b**, Left, the fraction of Arp2/3-mediated branches that survive over time. The dissociation of Arp2/3-mediated branches was observed and quantified in the presence of 0.1 μM full-length cortactin (olive green), or in the presence of 0.1 μM cortactin NtA (dark green for glutathione S-transferase (GST)-tagged NtA and light green for untagged NtA) or in the presence of 0.1 μM GST-N-WASP-CA (pink) in addition to 0.3 μM G-actin. The results of the control experiments with only 0.3 μM G-actin are shown in gray. Data for each curve were obtained from independent experiments. Right, schematic of actin-branch survival status in the assay (mother filament in dark gray, daughter filaments in light gray). **c**, Binding sites of cortactin NtA (left, green) and N-WASP-CA

(right, pink) on active Arp3 (orange) or inactive Arp3 (gray). Arp3 subdomains are numbered. **d**, Overlapping binding sites of cortactin NtA and class 1 NPF CA domain on Arp3 indicate how these proteins would compete for Arp3 binding. Active and inactive Arp3 structures are superposed by alignment of subdomains 3 and 4. Only a subset of Arp3 structural features are shown for clarity. **e**, Rotated view of overlaid active (orange) and inactive (gray) Arp3 structures with cortactin NtA and CA domain bound, as in **d**. Conformational differences of Arp3 α-helices at the cortactin NtA and CA binding sites in active or inactive Arp3 are indicated by an arrow and explain the sensitivities of these binding partners to the activation state of Arp3. The dashed lines in **d** and **e** indicate discontinuity in the N-WASP-CA domain structure.

daughter filament (Fig. 4a). In addition, they also reveal that the NtA indirectly stabilizes the branch by forming extensive interactions with activated Arp3 to promote DA1 D-loop insertion (Figs. 2a and 4a and Extended Data Fig. 5). The interactions between cortactin's NtA and Arp3 are specific to the activated conformation of Arp3, as computational docking of our NtA structure onto the inactive Arp3 conformation generates structural clashes (Fig. 2a and Extended Data Fig. 5e). To assess whether NtA alone can stabilize actin branches because of its preference for activated Arp3, we tested whether cortactin NtA,

in comparison to an actin-only control, could maintain branches in an in vitro debranching assay (Fig. 4b and Extended Data Fig. 7). NtA does provide protection. However, it was less effective than full-length cortactin, consistent with the notion that central-repeat binding maximizes branch stabilization.

## Discussion

Several studies have demonstrated the substantially higher affinity of cortactin for branch junctions than for unbranched filaments[22,50], but

have not provided direct evidence of the model of branch stabilization through mother-filament binding that is prevalent in the literature[2,22,27,51]. Structural cryo-EM is ideal for addressing this question, and has enabled us to visualize cortactin binding at the branch junction. Our data reveal exactly how cortactin supports Arp2/3-mediated stabilization of the actin branch: by binding to the active conformation of Arp3 and bridging across the longitudinal subunit interactions along the daughter filament.

Our observation that cortactin stabilizes activated Arp3 contrasts with destabilizing activities of VCA-containing class 1 NPFs and has implications for the coordinated regulation of actin branches. Our structure shows how binding of cortactin NtA to Arp3 would sterically block VCA binding sites—particularly of its C helix—to compete for Arp3 binding (Fig. 4c,d). VCA binding to Arp3 has been observed only on the inactive complex[17,20]; it does not readily associate with activated Arp3 in molecular dynamics simulations[52], and VCA binding would clash with the D-loops of incoming DA1 and DA2 (ref. 10). Consistent with a preference for inactivated Arp2/3, N-WASP-CA promotes branch destabilization in our debranching assay[49] (Fig. 4b and Extended Data Fig. 7). The overlapping binding sites of the cortactin NtA helix and the VCA helix are centered at the junction of Arp3 subdomains 1 and 3; the relative positions of these subdomains change upon Arp2/3 activation, and each protein is sensitive to these alterations (Fig. 4d).

Furthermore, the preferential binding of VCA and cortactin to inactive and activated Arp3, respectively, provides a mechanistic basis for the previously described displacement models of the synergistic promotion of Arp2/3 actin nucleation by VCA and cortactin[21,22,34]. VCA release from nascent actin branches is a necessary and rate-limiting step for branch formation and is accelerated by cortactin[21–23]. Our structure, further supported by a recent, related study[53], now shows that cortactin binding to Arp3 displaces the NPF CA domain, both by competition and because the activated Arp3 conformation favors NtA binding. The previously reported synergy of class 1 NPF VCA domains and cortactin at Arp2/3 branches therefore arises from VCA binding to and activating Arp2/3 followed by NtA accelerating VCA release and stabilizing the Arp2/3 activated state[21,22,34]. In addition, the observation that cortactin alone is only a weak activator of Arp2/3 nucleation has been puzzling and was thought to be because of its inability to recruit actin monomers to the nascent branch, which class 1 NPFs can do[32,51]. Our structure now shows that this weak stimulation of Arp2/3 nucleation is also because of the preference of the cortactin NtA for activated Arp3. This notion is consistent with the effects of cellular cortactin depletion, which indicate that the main role of cortactin is to stabilize Arp2/3-nucleated filaments, rather than as an activator of the complex per se[33,54].

It is also striking that our structure-based model reveals that the interaction mode of the 6.5 cortactin central repeats corresponds precisely to a half-turn of the F-actin helix. By contrast, the hematopoietic-cell-specific cortactin paralogue HS1 has only 3.5 repeats and would be predicted to interact only with DA1, DA3, DA5 and DA7, consistent with its lower affinity for F-actin[55]. Thus, although synergy in promoting branch formation and stabilization between cortactin and its relatives with class 1 NPFs is predicted to be conserved through the NtA, variations in the number of actin-binding repeats indicate how their regulated expression in different tissues could tune the local dynamics of branched actin networks.

In the Arp2/3 complex, not only does Arp3 form the structural template for the nucleated daughter filament, but its conformation favors binding partners such as cortactin and may also communicate to other cytoskeleton regulators, such as the debranching factor coronin, that the complex is activated[33]. Because actin-branch turnover is essential for the actin cytoskeleton to function normally, our visualization of cortactin has important implications for how it protects against debranching, whether through competition for Arp2/3 binding, protection of the daughter-filament junction or both[27,33,56–59].

This in turn could determine the extent to which Arp2/3 complexes remain bound to mother filaments following debranching and are thus available for further rounds of nucleation[60]. Our discovery of an α-helix in the cortactin NtA and characterization of its binding site at the junction of Arp3 subdomains 1 and 3 highlights the equivalence of this binding site to the binding cleft on actin, where a large number of actin-binding proteins interact and which also mediates longitudinal contacts in F-actin[44,45,61]. This emphasizes the conserved nature of the conformational changes that both Arp3 and actin undergo during actin nucleation and polymerization, and the importance of this hotspot in both proteins for binding regulators.

## Online content

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

## Methods

### Protein purification

Full-length mouse cortactin (residues 1–546, UniProt Q60598), human SPIN90 C terminus (residues 267–715, UniProt Q9NZQ3) and GST-tagged human N-WASP-VVCA and N-WASP-CA (residues 392–505 and 453–505, respectively, UniProt O00401) were purified following the protocol described by Cao et al.[49]. Cortactin NtA (residues 1–77) was purified using the same method as that used for full-length cortactin. Human Arp2/3 complex containing ArpC1B/C5L isoforms (UniProt P61160, P61158, O15143, O15144, O15145, P59998, Q9BPX5) was purified following the protocol described by Baldauf et al.[64].

Mouse capping protein α1β2 (UniProt P47753 and P47757-2) was co-expressed in BL21 Star DE3 cells using a pRSFDuet-1 plasmid with an N-terminal 6×histidine (6×His) tag fused to the α1 subunit. Cells were grown at 37 °C, and protein expression was induced with 20 µM IPTG when the optical density reached 1.1. After the addition of IPTG, cells were grown overnight at 16 °C. The next day, cells were collected by centrifugation at 5,000$g$ for 15 min, resuspended and lysed using a high-pressure homogenizer (Avesti Emulsiflex C3) in lysis buffer (50 mM Tris pH 8.0, 138 mM NaCl, 2.7 mM KCl with EDTA-free protease inhibitor (Roche)). The cell lysate was centrifuged at 49,500$g$ for 30 min to remove cell debris. The supernatant was transferred to a column with Ni-NTA Resin (Merck) and incubated for 1 h at 4 °C. The column was washed with lysis buffer and His-tagged capping protein dimer was eluted from the column in elution buffer (50 mM Tris pH 8.0, 138 mM NaCl, 2.7 mM KCl with 250 mM Imidazole). The eluted proteins were concentrated to 0.5 ml using Amicon Ultra-4 ml Centrifugal Filters (Millipore) and loaded onto a gel filtration column (Superdex 200 Increase 10/300 GL, GE Healthcare) on an ÄKTA system (GE Healthcare). The peak fractions containing capping protein were collected and buffer exchanged into a low-salt buffer (10 mM Tris pH 7.5, 10 mM KCl and 1 mM DTT). Finally, the proteins were loaded onto a 1 ml HiTrap Q HP column (GE Healthcare). Capping protein heterodimers were separated from other minor protein contaminants by linear gradient elution. The linear gradient was generated by combining high-salt buffer (10 mM Tris pH 7.5, 400 mM KCl and 1 mM DTT) with low-salt buffer (10 mM Tris pH 7.5, 10 mM KCl and 1 mM DTT). β/γ non-muscle actin from purified porcine brain was purchased from Hypermol (Cat 8401-01) and reconstituted with 200 µl ultrapure water to obtain a 1 mg ml$^{-1}$ solution in a buffer with 2 mM Tris-HCl pH 8.2, 2.0 mM ATP, 0.5 mM DTT, 0.1 mM CaCl$_2$, 1 mM NaN$_3$ and 0.2% disaccharides.

### Cryo-EM sample preparation

Branch reconstitution conditions were adapted from reports in the literature describing reconstitution of Arp2/3-complex-bound actin filaments[10,11,43]. Protein concentrations were optimized to enhance short-actin-branch formation and to minimize the preferred orientation problem caused by the 'Y'-shape of actin branches on the cryo-EM grid: (1) the actin concentration was kept low to prevent spontaneous nucleation and limit filament growth; (2) a high concentration of capping protein was added to limit daughter-filament growth. First, 1.7 µM Arp2/3, 1.7 µM VVCA, 16.1 µM SPIN90, 0.8 µM actin and 3.2 µM capping protein were mixed in 14.9 µl buffer containing 20 mM HEPES pH 7.5, 50 mM KCl, 1 mM EGTA, 1 mM MgCl$_2$, 0.2 mM ATP and 1 mM DTT and incubated at room temperature for 20 min. Then, 4.5 µl of 23.8 µM actin was added in nine separate additions, and the mixture was incubated at room temperature for 20 min; 1.2 µl of 80 µM capping protein was added in two separate additions with the third and seventh addition of actin. After the final addition of actin, 1.9 µM full-length cortactin was added, followed by another 20-min incubation. Finally, 10 µM phalloidin (Invitrogen) was added to stabilize the actin branches (summarized in Extended Data Fig. 1c).

Following incubation, 4 µl of the final reconstitution mix was applied to a glow-discharged C-flat 1.2/1.3 grid. The grid was plunge frozen using EM GP2 Automatic Plunge Freezer (Leica) with the following settings: sensor blotting, back blotting, additional movement of 0.3 mm, blotting time of 5 s, humidity of 98% and temperature of 22 °C.

### Cryo-EM data acquisition

Cryo-EM data (12,073 videos) were collected on a Titan Krios microscope (Thermo Fisher Scientific) operated at an accelerating voltage of 300 kV with a nominal magnification of ×81,000 and a pixel size of 1.067 Å. The data were collected with a K3 detector operating in super-resolution mode (bin2) with a BioQuantum energy filter (Gatan). Fifty frames for each micrograph were collected using EPU software with 14.8 e$^-$ pixel$^{-1}$s$^{-1}$ dose rate, an exposure time of 3.8 s, a total electron exposure dose of 49.4 e$^-$ per Å$^2$ and a defocus range of −0.9 to −2.4 µm.

### Cryo-EM data processing

Cryo-EM data were processed using CryoSPARC v3 (ref. 65). Movies were motion-corrected using Patch motion. Contrast transfer function (CTF) parameters were estimated using Patch CTF. We selected 8,518 micrographs with CTF fit resolution < 6.4 Å and total full-frame motion distance < 50 pixels for further data processing. Blob picker with a minimum diameter of 150 Å and a maximum diameter of 200 Å was used for particle picking, followed by particle extraction with a box size of 368 pixels and a binning factor of 4; 2,001,580 extracted particles were subjected to multiple rounds of two-dimensional (2D) classification to remove contaminants, carbon and non-branched portions of actin filaments. Class averages featuring various views of the actin branch junction were selected as templates for template picking. Then, 3,247,396 template-picked particles were subjected to multiple rounds of 2D classification. The sets of particles selected from the blob picker (167,590 particles) and template picker (244,162 particles) were subjected to ab initio reconstruction with two classes. After ab initio reconstruction, un-binned particles from these classes were re-extracted with a box size of 440, and each was subjected to homogeneous refinement with the best branch-like ab initio volume as the initial model. After homogeneous refinements and duplicate removal, the two stacks of particles were combined. The combined 179,923 particles were then subjected to a first round of non-uniform (NU)-refinement, followed by heterogeneous refinement with three classes to further classify particles. The volume of class 1 exhibited additional density on one side of the mother filament, and these particles were discarded. The remaining 130,915 particles from class 2 and class 3 were combined and subjected to a second round of NU refinement. Because all three classes differ only in the mother-filament region, we refer to the first NU-refinement reconstruction before heterogeneous refinement as the Arp2/3-daughter-filament consensus map. The second NU-refinement reconstruction is referred to as the mother-filament consensus map (Extended Data Fig. 2).

Local refinement with a mask around the mother filament on the mother-filament consensus map was used to improve the mother-filament density. Likewise, the Arp2/3-daughter filament consensus map was divided into three overlapping segments (the Arp2/3 complex, the daughter filament and the capping protein) and locally refined to improve the density of each segment. Before running local refinement on daughter filament and capping protein, the particles and consensus map were re-centered on DA3 using Volume Alignment Tools in cryoSPARC to improve the alignment because they are at the periphery of the consensus reconstruction. After local refinement on the daughter filament, the complete first cortactin F-actin repeat density was observed. After local refinement on the capping protein, the capping protein density was well resolved. Three-dimensional classification showed that the daughter-filament segments in all three classes shared an identical feature, containing only four actin subunits plus one capping protein heterodimer. A high molar ratio of capping protein to actin in our reaction mix contributes to the short daughter filament. The template picking and ab initio reconstruction step may

also introduce bias into the particle selection used in our reconstruction. Global resolution and local resolution of local refined maps were estimated in cryoSPARC (Extended Data Fig. 3).

## Model building

The four locally refined reconstructions were used to model Arp2/3 and cortactin NtA, the daughter filament and cortactin first central repeat, capping protein and the mother filament (Extended Data Figs. 2 and 3). Models of all seven Arp2/3 subunits, β-actin and capping protein created with the AlphaFold Monomer v2.0 pipeline were used initially[62,63]. They were rigidly fit into EM density using ChimeraX[66], followed by molecular-dynamics flexible fitting using ISOLDE[67]. Namdinator[68] was used to optimize bond geometry, and ISOLDE and Coot[69] were used at the end of the model-building process to manually fix Ramachandran outliers, rotamer outliers and clashes. AlphaFold predicted the N terminus of cortactin with low confidence except for one six-turn α-helix, corresponding to the α-helix in our EM density. The AlphaFold-predicted α-helix (residue 55–76) was well fitted into the EM density, with bulky side chains on one side of the α-helix facilitating its positioning. After the positioning of the NtA helix, the flanking cortactin residues (residues 21–54 and 80–116) were manually built using Coot[69].

## Structural analysis and visualization

Figures and videos of structures were made with ChimeraX[66]. Rise and twist angles shown in Extended Data Fig. 4 were calculated in PyMOL Molecular Graphics System, Version 2.5.4 (Schrödinger). The distances between interacting atoms in Extended Data Fig. 5 were measured in ChimeraX.

## Dissociation of branches by cortactin and CA motifs

Microfluidics experiments were conducted using poly-dimethyl-siloxane (PDMS, Sylgard) chambers with three inlets and one outlet, following the original protocol by Jegou et al.[70]. The microfluidic flows were monitored by a Microfluidic Flow Control System and Flow Units (Fluigent). Experiments were performed in buffer containing 5 mM Tris-HCl pH 7.0, 50 mM KCl, 1 mM $MgCl_2$, 0.2 mM EGTA, 0.2 mM ATP, 10 mM DTT, 1 mM DABCO and 0.1% BSA. The temperature was maintained at 25 °C by an objective heater (Oko-lab). Actin filaments were visualized using TIRF microscopy (Nikon TiE inverted microscope, iLAS2, Gataca Systems) equipped with a ×60 oil-immersion objective. Images were acquired using an Evolve EMCCD camera (Photometrics), controlled with the Metamorph software (version 7.10.4, from Molecular Devices).

Pointed-end anchored mother filaments (15% labeled with Alexa Fluor 488) and their branches (15% labeled with Alexa Fluor 568) were generated in a microfluidics chamber with a height of 20 μm and width of 1,600 μm, as described by Cao et al.[49]. During the experiment, actin branches were exposed to 0.3 μM actin as a control, or with an additional 0.1 μM cortactin, GST-N-WASP-VVCA or their mutants. The flow rate was set as high as 16 μl min$^{-1}$ while measurements were taken. The forces, ranging from 0.6 to 1 pN applied on the daughter filaments, were identical in each experiment. For each condition, the survival fraction of branches was quantified and plotted over time (Fig. 4b). For each condition, more than 40 branches were randomly chosen for further analysis, as previously reported by Cao et al.[49]. Each experiment was repeated independently three times, and all the repetitions were successful. The time points when half of the actin branches disappeared under different experimental conditions were plotted for comparison (Extended Data Fig. 7b). Fiji software was used to analyze images manually[71].

## Reporting summary

Further information on research design is available in the Nature Portfolio Reporting Summary linked to this article.

## Data availability

The cryo-EM reconstructions are deposited in the Electron Microscopy Data Bank under the following accession codes: daughter filament consensus reconstruction, EMDB-17553; Arp2/3 complex and cort-actin locally refined reconstruction, EMDB-17554; daughter filament and cortactin locally refined reconstruction, EMDB-17555; capping protein locally refined reconstruction, EMDB-17556; mother filament locally refined reconstruction, EMDB-17557; mother filament consensus reconstruction, EMDB-17558. The corresponding composite structural model is deposited in the Worldwide Protein Data Bank under the accession code PDB 8P94. PDB models used for structure comparison and model building are PDB 8E9B and PDB 6UHC. Source data are provided with this paper.

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

## Acknowledgements

This project has received funding from the European Research Council (ERC) under the European Union's Horizon 2020 research and innovation programme (grant agreement No 810207 to M.W. and C.A.M.). L.C. was supported by the European Union's Horizon 2020 Marie Sklodowka-Curie individual fellowship program (H2020-MSCA-IF-101028239 – MolecularArp). A.J. was supported by the ERC (grant StG-679116). M.W. is supported by the Francis Crick Institute, which receives its core funding from Cancer Research UK (CC2096), the UK Medical Research Council (CC2096) and the Wellcome Trust (CC2096). Cryo-EM data collected at the Institute of Structural and Molecular Biology (ISMB), Birkbeck, were on equipment funded by the Wellcome Trust (202679/Z/16/Z and 206166/Z/17/Z). We thank N. Lukoyanova and S. Chen for electron microscope support, D. Houldershaw for computing support at Birkbeck and G. Romet-Lemonne (Institut Jacques Monod), R. Treisman (the Francis Crick Institute), G. Waksman (Birkbeck, University of London), H Walden (University of Glasgow) and G. Zanetti (Birkbeck, University of London) for feedback on the manuscript.

## Author contributions

T.L. purified recombinant capping protein and conducted the cryo-EM sample preparation, data collection and structural analysis. L.C. purified all other recombinant Arp2/3 regulators and conducted and analyzed the debranching assay. M.M. prepared recombinant Arp2/3 complexes. A.J. provided GST-VVCA and actin for the debranching assay, and

technical advice on the debranching experiments. M.W. and C.A.M. supervised the project and wrote the paper, with input from all authors.

## Competing interests

The authors declare no competing interests.

## Additional information

**Extended data** is available for this paper at https://doi.org/10.1038/s41594-023-01205-2.

**Correspondence and requests for materials** should be addressed to Michael Way or Carolyn A. Moores.

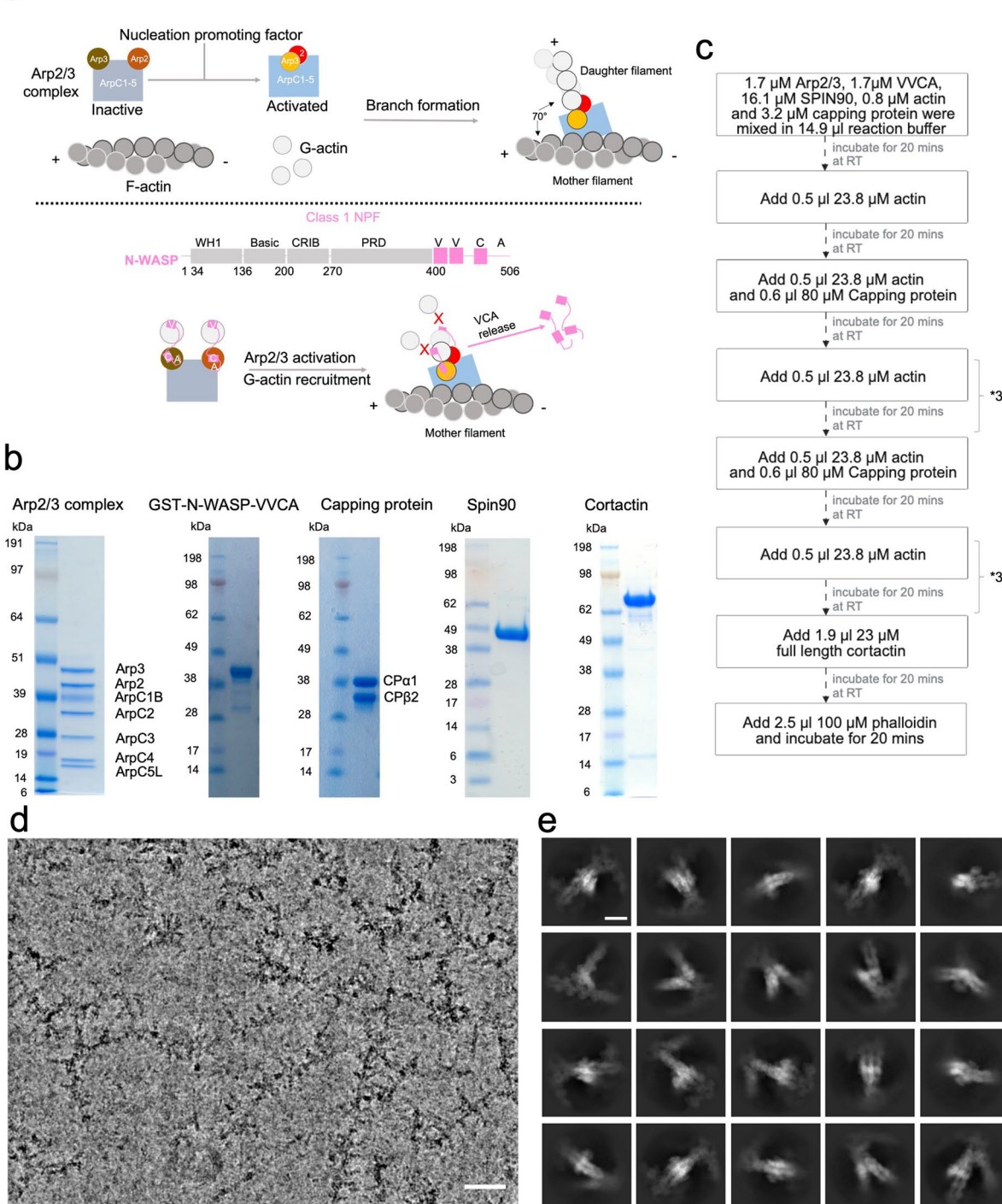

**Extended Data Fig. 1 | Overview of actin branch formation, purified proteins used in actin branch reconstitution and exemplar cryo-EM data. a)** Schematic of Arp2/3-mediated actin branch formation and role of Class 1 NPFs. When the Arp2/3 complex is activated by Class 1 NPFs, its Arp2 and Arp3 subunits rearrange into a short-pitch conformation, which acts as the template for daughter filament growth. The CA domain within the VCA domain of class 1 NPF interacts with Arp2/3 and V motif binds and recruits actin monomers. VCA must be released from the nascent branch junction prior to daughter filament elongation because it blocks the binding site for further daughter filament growth. **b)** SDS-PAGE gels showing purified proteins used in cryo-EM and microfluidics reconstitution experiments. Similar protein quantity and quality were obtained from at least 2 independent purification batches. **c)** Flow chart showing how the cryo-EM sample was prepared. **d)** A representative cryo-EM image of cortactin stabilized Arp2/3-mediated actin branches showing 'Christmas tree'-like mother filaments with multiple short daughter filaments extending from them. Scale bar = 50 nm. 8518 micrographs with similar image quality and branch density were collected and processed. **e)** Representative 2D class averages of particles selected using CryoSPARC blob picker and subjected to 2D classification showing multiple 2D projection views, which are used as templates for template picking. Scale bar = 10 nm.

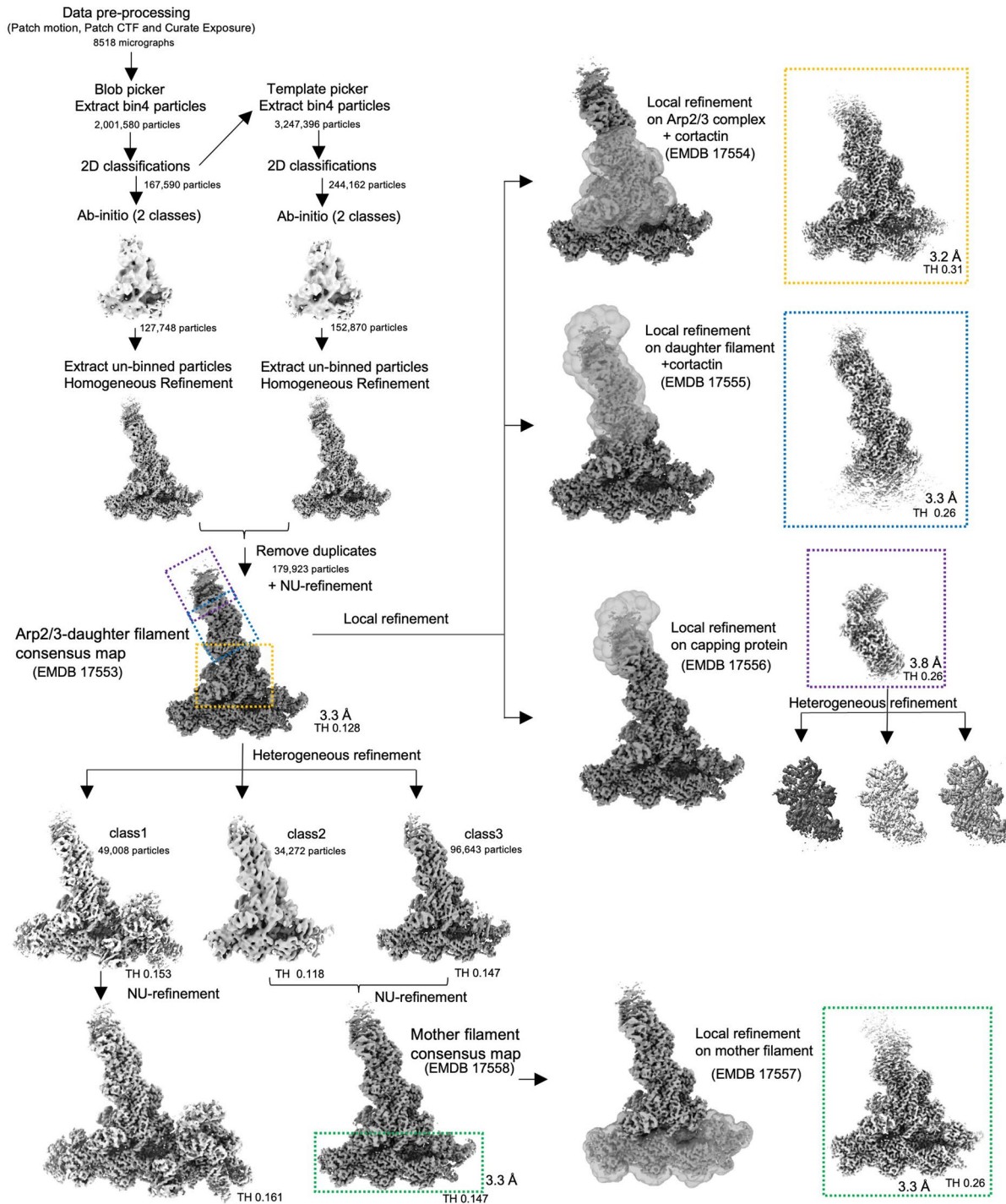

**Extended Data Fig. 2 | Image processing workflow for cryo-EM reconstruction.** The workflow used to generate the overlapping locally refined reconstructions of cortactin-bound Arp2/3 complex, daughter filament, capping protein and mother filament. Thresholds (THs) and global resolutions are indicated.

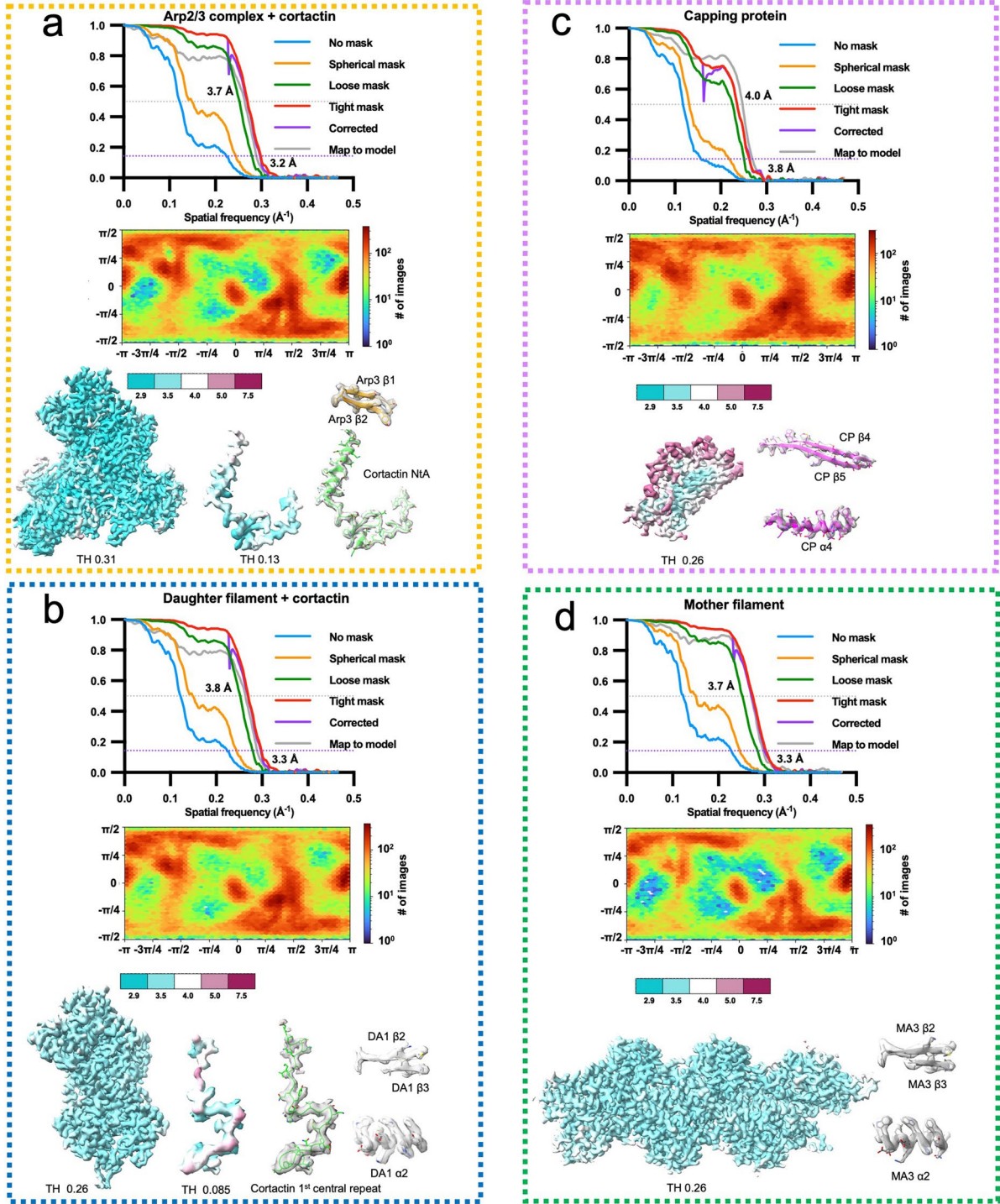

**Extended Data Fig. 3 | Cryo-EM data quality and validation of locally refined reconstructions.** For each locally refined reconstruction shown in Extended Data Fig. 2., half-map and map-model Fourier Shell Correlation (FSC), the angular distribution of particles used for 3D refinement, the well-resolved density used to generate the composite map colored by local resolution and representative regions of the density map with the final model are shown. Thresholds (THs) are indicated. FSC Cut-off 0.143 was used for half-map resolution estimation. FSC Cut-off 0.5 was used for map-model resolution estimation and local resolution estimation. (**a**) Locally refined reconstruction of cortactin-bound Arp2/3 complex. (**b**) Locally refined reconstruction of daughter filament. (**c**) Locally refined reconstruction of capping protein. (**d**) Locally refined reconstruction of mother filament.

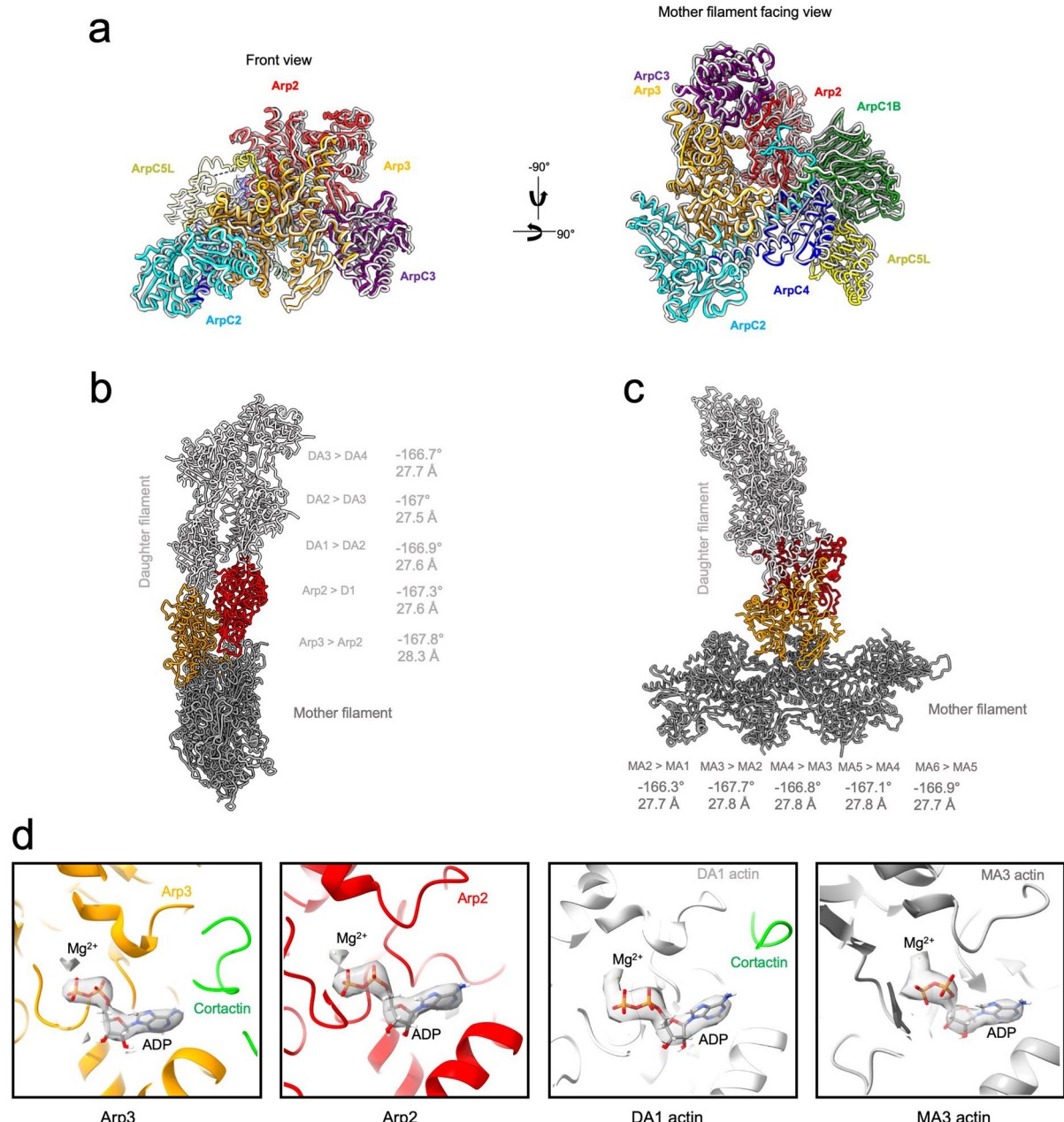

**Extended Data Fig. 4 | The activated Arp2/3 complex and ADP-F-actin in our cortactin-bound actin branch junction adopt canonical conformations.**
**a**) Structural alignment of cortactin-bound activated human Arp2/3 complex (this manuscript, subunits coloured) with activated bovine Arp2/3 complex (PDB 7tpt, subunits in light gray). Structures were aligned on ArpC2 (AA1-253) and Arp3 (AA1-36, AA60-153 and AA375-409). **b**) Arp2 and Arp3 in the activated Arp2/3 complex act as the template for daughter filament elongation. The

canonical rise and twist between daughter filament actin subunits (DA1 – DA4) and/or Arp subunits are indicated. **c**) The canonical rise and twist between mother filament actin subunits (MA1 – MA6) are indicated and show no evidence of distortion within the mother filament upon branch formation. **d**) Density (transparent) and models of ADPs (in stick representation) and Mg²⁺ (green dot) in Arp3, Arp2 and actin subunit DA1 in daughter and actin subunit MA3 in mother filament.

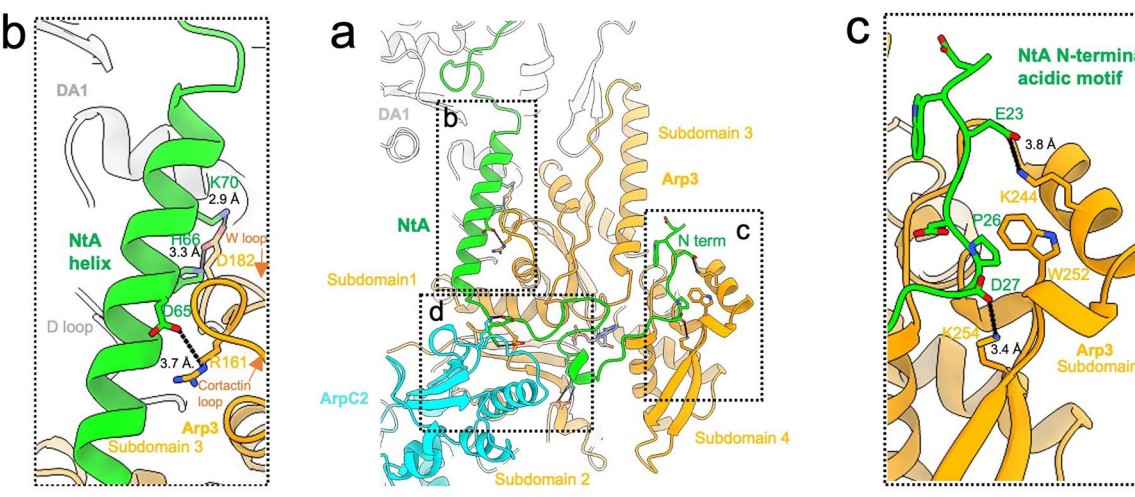

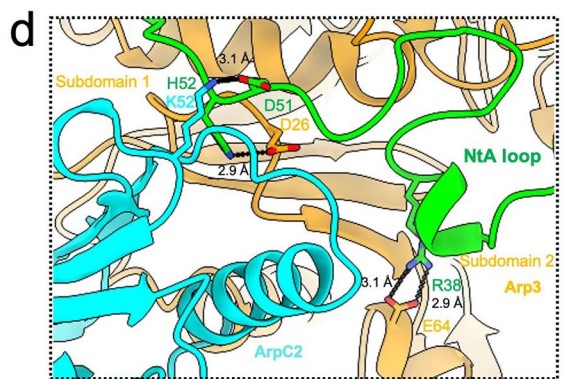

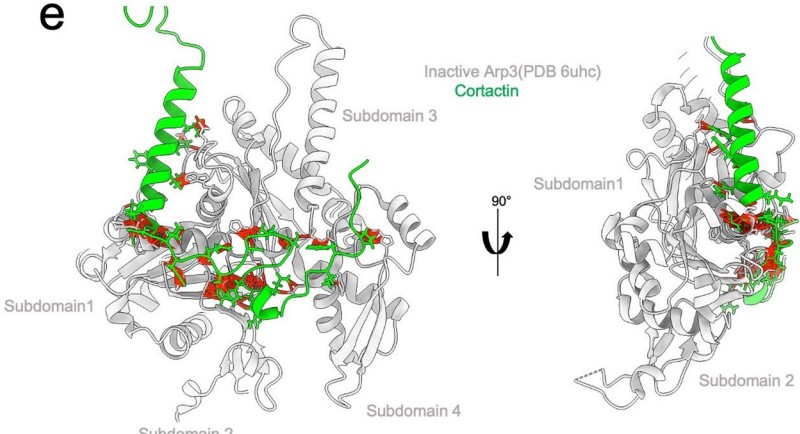

**Extended Data Fig. 5 | Interface details of the interactions between cortactin NtA and four subdomains of Arp3. a**) Overview of the cortactin NtA-Arp3-ARPC2 interface (similar to Fig. 2a). **b**) Details of the interaction between the cortactin NtA helix and Arp3 subdomain 3 (W loop and cortactin loop) of Arp3. **c**) Details of the interactions between NtA loop and subdomain 4. **d**) Details of the interaction between NtA loop and subdomain 1 and 2 of Arp3 and ArpC2. **e**) Computational docking of our NtA structure onto the inactive Arp3 conformation generates structural clashes. Inactive Arp3 (from PDB 6uhc) is positioned by aligning on subdomains 3 and 4 of our activated Arp3 structure. Atom pairs with van der Waals overlap ≥ 0.7 Å (after subtracting 0.4 Å for H-bonding) were classified as clashes.

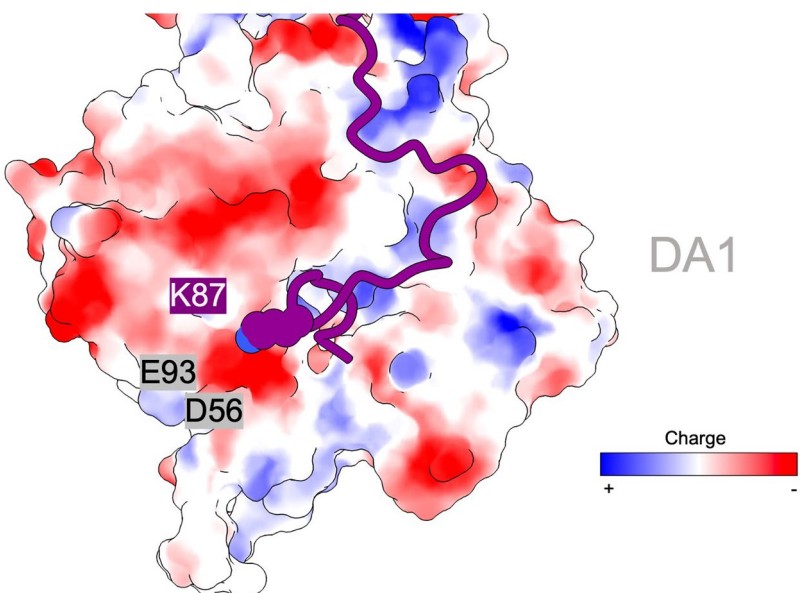

**Extended Data Fig. 6 | Lysine 87 acetylation in the first central repeat of cortactin is predicted to affect electrostatic interactions with the first daughter filament actin subunit.** Cortactin K87 is in close proximity to a negatively charged patch formed by E93 and D56 in DA1. K87 is shown as sphere representation. DA1 is shown in surface representation coloured by electrostatic potential.

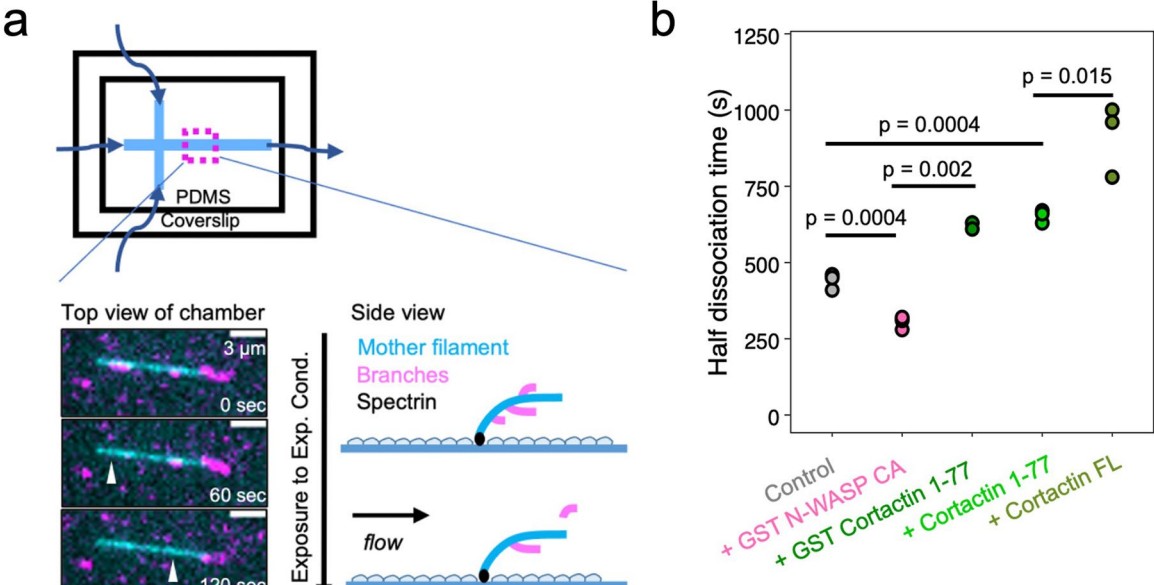

**Extended Data Fig. 7 | Microfluidics-based debranching assay shows the role of cortactin NtA in branch stabilization. a**) A microfluidics setup was used to study the stability of Arp2/3-mediated branches. Actin filaments were attached to the surface of coverslip via their pointed ends. Actin branches were generated on top of the pre-existing filament using differently labelled actin. The dissociation of actin branches under the experimental conditions were observed and quantified. **b**) The time point when half of the actin branches have dissociated under different experimental conditions, taken from the same experiments as depicted in Fig. 4b. Each point represents the half dissociation time in an independent experiment. Each experiment was repeated three times independently and all the repeats were successful. The p-value is estimated by two tailed unpaired t-test.

# Reporting Summary

## Statistics

For all statistical analyses, confirm that the following items are present in the figure legend, table legend, main text, or Methods section.

| n/a | Confirmed | |
|---|---|---|
| ☐ | ☒ | The exact sample size (*n*) for each experimental group/condition, given as a discrete number and unit of measurement |
| ☐ | ☒ | A statement on whether measurements were taken from distinct samples or whether the same sample was measured repeatedly |
| ☐ | ☒ | The statistical test(s) used AND whether they are one- or two-sided *Only common tests should be described solely by name; describe more complex techniques in the Methods section.* |
| ☒ | ☐ | A description of all covariates tested |
| ☒ | ☐ | A description of any assumptions or corrections, such as tests of normality and adjustment for multiple comparisons |
| ☐ | ☒ | A full description of the statistical parameters including central tendency (e.g. means) or other basic estimates (e.g. regression coefficient) AND variation (e.g. standard deviation) or associated estimates of uncertainty (e.g. confidence intervals) |
| ☐ | ☒ | For null hypothesis testing, the test statistic (e.g. *F*, *t*, *r*) with confidence intervals, effect sizes, degrees of freedom and *P* value noted *Give P values as exact values whenever suitable.* |
| ☒ | ☐ | For Bayesian analysis, information on the choice of priors and Markov chain Monte Carlo settings |
| ☒ | ☐ | For hierarchical and complex designs, identification of the appropriate level for tests and full reporting of outcomes |
| ☒ | ☐ | Estimates of effect sizes (e.g. Cohen's *d*, Pearson's *r*), indicating how they were calculated |

*Our web collection on statistics for biologists contains articles on many of the points above.*

## Software and code

Policy information about availability of computer code

| Data collection | Cryo-EM dataset of cortactin stabilized actin branch junction was collected on Titan Krios with EPU software (version 2.14, Thermo Fisher Scientific). The data were collected with a K3 detector operating in super-resolution mode (bin2) with a BioQuantum energy filter (Gatan).The debranching essays were performed with TIRF microscopy (Nikon TiE inverted microscope, iLAS2, Gataca Systems) equipped with a 60× oil-immersion objective. Images were acquired using an Evolve EMCCD camera (Photometrics), controlled with the Metamorph software (version 7.10.4, from Molecular Devices). |
|---|---|
| Data analysis | Cryo-EM data was processed using CryoSPARC v3.Model was built using AlphaFold Monomer v2.0 pipeline,ISOLDE,Namdinator and Coot. Rise and twist angles of actin and actin related proteins were calculated in PyMOL Molecular Graphics System, Version 2.5.4 Schrödinger, LLC. The distance between interacting atoms were measured in ChimeraX. Structural figures and movies were made with ChimeraX. The debranching rate was analyzed with Fiji (version 2.14.0/1.54f). |

For manuscripts utilizing custom algorithms or software that are central to the research but not yet described in published literature, software must be made available to editors and reviewers. We strongly encourage code deposition in a community repository (e.g. GitHub). See the Nature Portfolio guidelines for submitting code & software for further information.

## Data

Policy information about [availability of data](availability of data)

All manuscripts must include a [data availability statement](data availability statement). This statement should provide the following information, where applicable:

- Accession codes, unique identifiers, or web links for publicly available datasets
- A description of any restrictions on data availability
- For clinical datasets or third party data, please ensure that the statement adheres to our [policy](policy)

> The cryo-EM reconstructions are deposited in the Electron Microscopy Data Bank under the following accession codes: Daughter filament consensus reconstruction - EMDB-17553; Arp2/3 complex and cortactin locallyl refined reconstruction – EMDB-17554; Daughter filament and cortactin locally refined reconstruction – EMDB-17555; Capping protein locally refined reconstruction – EMDB-17556; Mother filament locally refined reconstruction – EMDB-17557; mother filament consensus reconstruction – EMDB-17558. The corresponding composite structural model is deposited in the Worldwide Protein Data Bank under the accession code PDB ID: 8P94. PDB models used for structure comparison and model building are PDB ID: 8E9B and PDB ID: 6UHC. Source data are provided with this paper.

## Research involving human participants, their data, or biological material

Policy information about [studies with human participants or human data](studies with human participants or human data). See also policy information about [sex, gender (identity/presentation), and sexual orientation](sex, gender (identity/presentation), and sexual orientation) and [race, ethnicity and racism](race, ethnicity and racism).

| | |
|---|---|
| Reporting on sex and gender | n/a |
| Reporting on race, ethnicity, or other socially relevant groupings | n/a |
| Population characteristics | n/a |
| Recruitment | n/a |
| Ethics oversight | n/a |

Note that full information on the approval of the study protocol must also be provided in the manuscript.

# Field-specific reporting

Please select the one below that is the best fit for your research. If you are not sure, read the appropriate sections before making your selection.

☒ Life sciences ☐ Behavioural & social sciences ☐ Ecological, evolutionary & environmental sciences

For a reference copy of the document with all sections, see [nature.com/documents/nr-reporting-summary-flat.pdf](nature.com/documents/nr-reporting-summary-flat.pdf)

# Life sciences study design

All studies must disclose on these points even when the disclosure is negative.

| | |
|---|---|
| Sample size | Cryo-EM sample size (12,073 movies) was determined based on previous similar studies (Ding et al.2022, PMID 35622886; Shaaban et al. 2020, 32839613), by microscope availability and data quality. After manually curating collected movies, 8,518 micrographs with CTF fit resolution < 6.4 Å and total full-frame motion distance < 50 pixels were selected for further analysis. For debranching assays, > 40 branches were chosen randomly for each condition and according to previously published analysis (Cao et al, 2023, PMID 36939020), which produces a survival curve of the branches that can be fit with an exponential equation and which is determined by branch disassociation rate. |
| Data exclusions | DATA EXCLUSIONS: Cryo-EM images were selected in a non-biased manner using well defined criteria (CTF fit resolution, total full-frame motion distance etc). For debranching assays, actin branches which are obviously abnormal, for example, sticking to the cover-slip surface, were excluded. |
| Replication | REPLICATION: For the cryo-EM dataset, similar images were obtained from 2-3 preliminary test dataset. Data for 3D reconstruction were collected from one grid. For debranching assays, each experiment was repeated three times independently and all replication attempts were successful. |
| Randomization | RANDOMIZATION: During cryoEM data processing, each dataset was randomly split in two for calculating gold-standard Fourier Shell Correlation (FSC). For debranching assays, all branches were generated under the same condition and subsequently exposed to different experimental variables. Other variables such as temperature and tension on the filaments were kept constant. Actin branches were randomly picked at time zero for analysis. |
| Blinding | BLINDING: Blinding is not a common practice in cryo-EM where the experience of the researcher on sample behaviour will benefit the efficiency and accuracy of data collection. During data processing, the data were mostly automatically processed using unbiased software. Given the complex and flexible nature of our sample, researchers need to select the targeted 2D and 3D classes in an unblinded way for further data processing. Rigorous checking for model bias is performed at multiple stages. For debranching assays, once the imaging field of view is chosen, the imaging took place automatically. The investigator cannot manipulate data acquisition based on the experimental condition. |

# Reporting for specific materials, systems and methods

We require information from authors about some types of materials, experimental systems and methods used in many studies. Here, indicate whether each material, system or method listed is relevant to your study. If you are not sure if a list item applies to your research, read the appropriate section before selecting a response.

## Materials & experimental systems

| n/a | Involved in the study |
|-----|----------------------|
| ☒ | Antibodies |
| ☒ | Eukaryotic cell lines |
| ☒ | Palaeontology and archaeology |
| ☒ | Animals and other organisms |
| ☒ | Clinical data |
| ☒ | Dual use research of concern |
| ☒ | Plants |

## Methods

| n/a | Involved in the study |
|-----|----------------------|
| ☒ | ChIP-seq |
| ☒ | Flow cytometry |
| ☒ | MRI-based neuroimaging |

