## [Peer Review File · Nature Structural & Molecular Biology]

Peer Review Information

Manuscript Title: Cortactin stabilizes actin branches by bridging activated Arp2/3 to its nucleated actin filament

Corresponding author name(s): Michael Way, Carolyn Moores

Reviewer Comments & Decisions:

Decision Letter, initial version:

Message: 12th Sep 2023

Dear Dr. Moores,

Thank you again for submitting your manuscript "Cortactin stabilizes actin branches by bridging activated Arp2/3 to its nucleated actin filament". I apologize for the delay in responding, which resulted from the difficulty in obtaining suitable referee reports. Nevertheless, we now have comments (below) from the 3 reviewers who evaluated your paper. In light of those reports, we remain interested in your study and would like to see your response to the comments of the referees, in the form of a revised manuscript.

You will see that while reviewers appreciate the results, they raise several concerns which will need to be addressed in a revision. Specifically, we would ask you to consider expanding the discussion on the ability of cortactin to activate Arp2/3, as suggested by reviewer #1. In line with reviewer's #2 comments we would ask that you provide more details regarding the methods and the constructs used in the study. We agree with reviewer #3 that the notions of preferential affinity of cortactin towards active Arp2/3 should be addressed or at least further discussed, as well as the points about synergy and competition between cortactin and other NPFs. Furthermore, we would encourage further discussion of points which contradict previous biochemical data suggesting cortactin binding to mother filament.

Please be sure to address/respond to all concerns of the referees in full in a point-by-point response and highlight all changes in the revised manuscript text file. If you have comments that are intended for editors only, please include those in a separate cover letter.

We are committed to providing a fair and constructive peer-review process. Do not hesitate to contact us if there are specific requests from the reviewers that you believe are

technically impossible or unlikely to yield a meaningful outcome.

We expect to see your revised manuscript within 10 weeks. If you cannot send it within this time, please contact us to discuss an extension; we would still consider your revision, provided that no similar work has been accepted for publication at NSMB or published elsewhere.

Reporting Summary:

When submitting the revised version of your manuscript, please pay close attention to our [href="https://www.nature.com/nature-portfolio/editorial-policies/image-integrity">Digital Image Integrity Guidelines. and to the following points below:](https://www.nature.com/nature-portfolio/editorial-policies/image-integrity)

Please note that all key data shown in the main figures as cropped gels or blots must be presented in uncropped form, with molecular weight markers. These data can be aggregated into a single supplementary figure item. While these data can be displayed in a relatively informal style, they must refer back to the relevant figures.

SOURCE DATA: we urge authors to provide, in tabular form, the data underlying the graphical representations used in figures. This is to further increase transparency in data reporting, as detailed in this editorial (<http://www.nature.com/nsmb/journal/v22/n10/full/nsmb.3110.html>). Spreadsheets can be submitted in excel format. Only one (1) file per figure is permitted; thus, for multi-paneled figures, the source data for each panel should be clearly labeled in the Excel file;

alternately the data can be provided as multiple, clearly labeled sheets in an Excel file. When submitting files, the title field should indicate which figure the source data pertains to.

Data availability: this journal strongly supports public availability of data. All data used in accepted papers should be available via a public data repository, or alternatively, as Supplementary Information. If data can only be shared on request, please explain why in your Data Availability Statement, and also in the correspondence with your editor. Please note that for some data types, deposition in a public repository is mandatory - more information on our data deposition policies and available repositories can be found below: <https://www.nature.com/nature-research/editorial-policies/reporting-standards#availability-of-data>

[Redacted]

Sincerely,

Katarzyna Ciazynska
(she/her)
Associate Editor
Nature Structural & Molecular Biology
<https://orcid.org/0000-0002-9899-2428>

Referee expertise:

Referee #1: cytoskeleton

Referee #2: cytoskeleton, cell biology

Referee #3: cryo-EM, actin

Reviewers' Comments:

Reviewer #1:

Remarks to the Author:

Liu et al. provide a very comprehensive study on the precise mode of interaction of cortactin, in particular its N-terminus and the first F-actin binding repeat with the Arp2/3 complex containing branch structure and the nascent, i.e. daughter filament. Most of the data were generated by cryo-EM, but there are also a few debranching assays using microfluidics devices (Fig. 4b and extended data Fig. 6) in order to explore differential effects on branch stabilization by cortactin fragments versus full length. Due to the extremely short branches formed (effected by heterodimeric CP), I assume testing larger cortactin constructs would not have made so much sense, so precise, continuous binding of additional cortactin F-actin binding repeats (up to 6.5) was deduced based on sequence information and structural modelling.

I believe this manuscript shows unprecedented detail and hence mechanistic understanding of cortactin binding to active Arp3 and the nascent daughter filament, so will be very important and interesting for a quite broad community interested in Arp2/3-dependent actin remodeling and cortactin biology. I am very excited about the results, which are presented in a very clear and educative fashion, so I have no objections to publishing this rapidly!

Instead, I am just providing a few very minor suggestions for potential text amendments, which are up to the authors whether they would like to consider them or not:

1) I believe the community has suffered from the persistent view that cortactin constitutes an activator of Arp2/3 complex-dependent actin remodeling, even though it is mentioned everywhere (including the present paper) that the extent of activation as compared to canonical, class 1 nucleation promoting factors (NPFs) is comparably weak. I believe the authors describe here how such apparent, weak activation can be caused in vitro, which will be likely due to the favored interaction of cortactin with active Arp3 plus branch and daughter filament stabilization, but whether those interactions still justify to speak of an ACTIVATOR of Arp2/3-mediated actin assembly, is questionable to me. I am specifically referring to the sentence in lines 167-9 (page 6), in which the authors word: "Further, the preferential binding of VCA and cortactin to inactive and activated Arp3 respectively provides a mechanistic basis for displacement models of the SYNERGISTIC ACTIVATION of

Arp2/3 by VCA and cortactin”, which seems a bit misleading in this context. I understand that this might appear as sole semantics, but large parts of the community still consider cortactin an essential Arp2/3 activator in vivo. This is largely based on comparably old but admittedly relevant literature, such as Uruno et al., 2021 (doi: 10.1038/350600519) or Weaver et al., 2001 (doi: 10.1016/s0960-9822(01)00098-7), which are both cited.

However, the fact that cortactin-deficient lamellipodia (Lai et al., 2009, doi: 10.1091/mbc.e08-12-1180) and Vaccinia tails (Abella et al., doi:10.1038/ncb3286) show defects in stability perhaps, but not in the rates of Arp2/3 complex turnover and hence activation, is not mentioned. Based on these thoughts, the authors might consider revising their statements on “synergistic activation of Arp2/3 by VCA and cortactin”.

2) The authors nicely explain in the manuscript text (e.g. line 62) that cortactin is considered a class 2 NPF, because it does not bind actin monomers through V or W-domains (like class 1 NPFs such as n-WASP), but instead harbors an F-actin binding domain etc, which is all fine. However, the abstract (I understand for the sake of brevity) is not consistent with this view, as it talks about cortactin binding “activated Arp3 in contrast to other NPFs” on one hand (line 36), but towards its end, uses wording that implies cortactin not to belong to the group of NPFs at all. This is because the authors word: “cortactin binding to Arp3 is incompatible with NPF interaction” (lines 39/40), which does not make sense, of course, if it constitutes an NPF itself. And in the last sentence, the authors state: “Our data have uncovered why cortactin displaces NPFs”, which is problematic for the same reason. So I would recommend solving these wording discrepancies.

3) Very minor point: I guess in case of N-WASP, it would be more precise to call the C-terminal activation fragment VVCA (instead of “VCA”) in extended data Fig. 1B and throughout the text. This would be consistent with the domain shown in extended data Fig. 1A and the fact that the fragment comprises aa392-505 of human N-WASP (line 228 in the Methods), and because the term VCA could be misunderstood as a shorter fragment just harboring the C-terminal WH2-domain, which would not be correct.

Reviewer #2:

Remarks to the Author:

Summary and major comments:

This paper describes a cryo-EM structure of an N-terminal fragment of Class 2 nucleation promoting factor, Cortactin (CTTN), bound to Arp2/3 complex on an actin branch. The structure provides valuable insights into how CTTN stabilizes actin branches as it shows that the CTTN Nta domain stabilizes Arp3 in an activated state, whereas the first central repeat domain of CTTN binds and stabilizes the daughter filament. Whereas the claim of the authors that this is how CTTN stabilizes branches is valid, their claim that the central repeat domains “exclusively” (Line 138) bind to the daughter filament is interpolated from interaction of the first central domain of CTTN with the daughter filament and a model based on that interaction. Given that full length CTTN has been shown to bundle actin filaments and branches (Helgeson et al., 2018), this raises the question whether some of the more C-terminal repeat domains in CTTN could cross over to the mother filament. As such, their structure does not support this claim and it should be corrected.

Additionally, this structure is an adaptation of two recent papers that reconstructed cryo-EM structures of Arp2/3 on an actin branch (Chou et al., 2022; Ding et al., 2022). This makes that the binding of CTTN to the branch is the main novelty of the presented structure. Because of this, it is a pity that the authors were not able to resolve binding of additional CTTN repeat domains, which would have increased the impact of the structure.

Major comments:

1. Based on the methods and Fig 1, it is unclear whether the authors attempted to obtain a structure with full length CTTN and that only the Nta and first central repeat domain could be resolved or that only the N-terminal fragment was used to begin with. The authors should state clearly in the manuscript, Methods and Fig 1 whether the structure they obtained was reconstructed with CTTN 1-177 only or with full length CTTN. In case of the latter, they should mention that they were only able to resolve part of the full length protein. Also, is this partial CTTN structure the result of the length of the branches that were obtained because of the high capping protein concentration?
2. To point 1, the authors deviated from the method established in Chou et al. (2022) to assemble branches and included SPIN90. Can the SPIN90 bound particles be identified and if so, can more CTTN central repeats be resolved? Why was SPIN90 included?
3. Fig 3: The authors should also include the actin residues that constitute the other side of the hydrophilic and hydrophobic contacts at the CTTN-F-actin interface.
4. Fig 3d: Please indicate the lysines that have been reported to undergo acetylation and show how they interact with F-actin.

Minor concerns:

1. The authors should include a flow chart in Fig 1, showing how the EM samples were prepared, including all the other proteins that were used to preassemble branches.
2. In Fig 1 and 2, the color of CTTN is referred to as "lime", but looks more like green. It would be better to just call it "green".
3. Fig 3d: The numbers in the cyan-colored repeat should be indicated in white so they are visible. Also, the CTTN repeat 3 color is different from the model shown in 3c. Please use consistent color coding.
4. Following the flow of the manuscript, Fig 4 A-B would fit better into Fig 3 than Fig. 4.

Reviewer #3:

Remarks to the Author:

This manuscript by Liu et al., for the first time reports a high resolution structure of cortactin stabilized Arp2/3 complex mediated branched actin junction. The structure shows cortactin bound to daughter filament, but not to the mother filament. This is contrary to what was believed in the field. However, this unique observation sheds light into how cortactin stabilizes these branched actin junctions and possibly WDS-Arp2/3 nucleated linear actin filaments. Furthermore, the structure also reveals at unprecedented details the molecular interactions between the NtA of cortactin and Arp3 subunit. Additionally, the structure reveals that cortactin preferentially interacts with Arp3 and not with Arp2 and elucidates the competitive binding of cortactin with Type-I NPF. Besides the NtA region of cortactin, in the structure the first actin binding repeat of cortactin was well resolved. The remaining 5.5 repeats were not resolved in the structure possibly due to the short length of the daughter filament. Overall this a well put together structural study which provides new insight into what was recently shown by Cao et al., in EMBO J. 2023.

Here are some minor comments and suggestions about the manuscript:

1. It is unclear as to how the authors claim that there is preferential affinity of cortactin Nta towards "activated" Arp3? Does that mean that Cortactin has lower binding affinity to inactive Arp2/3 complex? If so, then how did the authors come to this conclusion?
2. What could be the reason why the authors failed to observe any cortactin bound to the mother filament, while previous biochemical data showed binding of cortactin to mother

actin filaments?

3. It would be better if the authors also provide some explanation about synergy between cortactin and other NPFs, while also competing with them.

Overall it is a nice structural study that is of relevance to the field of branched actin nucleation and stability and must be published

Author Rebuttal to Initial comments

Reviewer #1

Liu et al. provide a very comprehensive study on the precise mode of interaction of cortactin, in particular its N-terminus and the first F-actin binding repeat with the Arp2/3 complex containing branch structure and the nascent, i.e. daughter filament. Most of the data were generated by cryo-EM, but there are also a few debranching assays using microfluidics devices (Fig. 4b and extended data Fig. 6) in order to explore differential effects on branch stabilization by cortactin fragments versus full length. Due to the extremely short branches formed (effected by heterodimeric CP), I assume testing larger cortactin constructs would not have made so much sense, so precise, continuous binding of additional cortactin F-actin binding repeats (up to 6.5) was deduced based on sequence information and structural modelling.

I believe this manuscript shows unprecedented detail and hence mechanistic understanding of cortactin binding to active Arp3 and the nascent daughter filament, so will be very important and interesting for a quite broad community interested in Arp2/3-dependent actin remodelling and cortactin biology. I am very excited about the results, which are presented in a very clear and educative fashion, so I have no objectives to publishing this rapidly!

Instead, I am just providing a few very minor suggestions for potential text amendments, which are up to the authors whether they would like to consider them or not:

1) I believe the community has suffered from the persistent view that cortactin constitutes an activator of Arp2/3 complex-dependent actin remodeling, even though it is mentioned everywhere (including the present paper) that the extent of activation as compared to canonical, class 1 nucleation promoting factors (NPFs) is comparably weak. I believe the authors describe here how such apparent, weak activation can be caused in vitro, which will be likely due to the favored interaction of cortactin with active Arp3 plus branch and daughter filament stabilization, but whether those interactions still justify to speak of an ACTIVATOR of Arp2/3-mediated actin assembly, is questionable to me. I am specifically referring to the sentence in lines 167-9 (page 6), in which the authors word: "Further, the preferential binding of VCA and cortactin to inactive and activated Arp3 respectively provides a mechanistic basis for displacement models of the SYNERGISTIC ACTIVATION of Arp2/3 by VCA and cortactin", which seems a bit misleading in this context. I understand that this might appear as sole semantics, but large parts of the community still consider cortactin an essential Arp2/3 activator in vivo. This is largely based on comparably old but admittedly relevant literature, such as Uruno et al., 2021 (doi: 10.1038/350600519) or Weaver et al., 2001 (doi: 10.1016/s0960-9822(01)00098-7), which are both cited. However, the fact that cortactin-deficient lamellipodia (Lai et al., 2009, doi: 10.1091/mbc.e08-12-1180) and Vaccinia tails (Abella et al., doi:10.1038/ncb3286) show defects in stability perhaps, but not in the rates of Arp2/3 complex turnover and hence activation, is not mentioned. Based on these thoughts, the authors might consider revising their statements on "synergistic activation of Arp2/3 by VCA and cortactin".

1) We agree with the reviewer's analysis. Based on the data available to date, it has been plausible for the community to consider cortactin a (very) weak Arp2/3 activator compared to VCA. However, as the reviewer points out, one of the exciting aspects of our structure is that it sheds new light on cortactin's properties and recasts it as a selective stabilizer of activated Arp3 rather than an activator. We take the reviewer's

point about the mentioned text being misleading in this context, and have adjusted it to read: "Further, the preferential binding of VCA and cortactin to inactive and activated Arp3 respectively provides a mechanistic basis for the previously described displacement models of the synergistic promotion of Arp2/3 actin nucleation by VCA and cortactin." (lines 175-178). We have also tried to refocus attention on this stabilizing role of cortactin in new text (lines 188-191), incorporating the mentioned references.

2) The authors nicely explain in the manuscript text (e.g. line 62) that cortactin is considered a class 2 NPF, because it does not bind actin monomers through V or W-domains (like class 1 NPFs such as n-WASP), but instead harbors an F-actin binding domain etc, which is all fine. However, the abstract (I understand for the sake of brevity) is not consistent with this view, as it talks about cortactin binding "activated Arp3 in contrast to other NPFs" on one hand (line 36), but towards its end, uses wording that implies cortactin not to belong to the group of NPFs at all. This is because the authors word: "cortactin binding to Arp3 is incompatible with NPF interaction" (lines 39/40), which does not make sense, of course, if it constitutes an NPF itself. And in the last sentence, the authors state: "Our data have uncovered why cortactin displaces NPFs", which is problematic for the same reason. So I would recommend solving these wording discrepancies.

2) Thanks a lot to the reviewer for flagging these inconsistencies. We have adjusted the abstract text in the places mentioned to ensure consistency.

3) Very minor point: I guess in case of N-WASP, it would be more precise to call the C-terminal activation fragment VVCA (instead of "VCA") in extended data Fig. 1B and throughout the text. This would be consistent with the domain shown in extended data Fig. 1A and the fact that the fragment comprises aa392-505 of human N-WASP (line 228 in the Methods), and because the term VCA could be misunderstood as a shorter fragment just harboring the C-terminal WH2-domain, which would not be correct.

3) The reviewer is absolutely correct and we have adjusted the Methods text and Extended Data Fig. 1b accordingly, and included VVCA in the newly added Extended Data Fig. 1c (Reviewer response point 11). However, for clarity, when referring to VCA-containing NPFs more generally (e.g. paragraph beginning line 163), we have retained the generic term VCA.

Reviewer #2:

Summary and major comments:

This paper describes a cryo-EM structure of an N-terminal fragment of Class 2 nucleation promoting factor, Cortactin (CTTN), bound to Arp2/3 complex on an actin branch. The structure provides valuable insights into how CTTN stabilizes actin branches as it shows that the CTTN Nta domain stabilizes Arp3 in an activated state, whereas the first central repeat domain of CTTN binds and stabilizes the daughter filament. Whereas the claim of the authors that this is how CTTN stabilizes branches is valid, their claim that the central repeat domains "exclusively" (Line 138) bind to the daughter filament is interpolated from interaction of the first central domain of CTTN with the daughter filament and a model based on that interaction. Given that full length CTTN has been shown to bundle actin filaments and branches (Helgeson et al., 2018), this raises the question whether some of the more C-terminal repeat

domains in CTTN could cross over to the mother filament. As such, their structure does not support this claim and it should be corrected.

4) The reviewer is probably referring to Helgeson et al (2014) PMID: 25160634 (Helgeson et al (2018) PMID: 29487209 is a study of kinetochore microtubules), where bundling of actin branches in the presence of full length cortactin was indeed reported. However, that study did not directly show that the cortactin inducing bundles was also bound to Arp3 at the branch. Indeed, the representative image in Fig. 3A in that paper shows bundling of adjacent branched filaments $\sim 1\mu\text{m}$ from the branches (which themselves remain distinctly separated), whereas our model suggests that the 5.5 repeats of cortactin not visualized in our reconstruction would only stretch $\sim 0.03\mu\text{m}$ while bound at the branch.

Given its multivalency and apparent flexibility, we cannot absolutely exclude that cortactin could reach back to bind the mother filament in a sample such as ours with short daughter filaments. However, we think this is unlikely because:

- there is no evidence of the filament bundling that might be predicted on this basis in our cryo-EM sample (see e.g. Extended Data Fig. 1d and 1e)
- we have not found evidence of additional density along the mother filament reconstruction that could correspond to any cortactin binding like this (Rebuttal Figure 1, below).

Rebuttal Figure 1. Difference map calculations show no evidence of cortactin mother filament binding.

- a) Views of mother filament local refined reconstruction with mother filament subunits MA1 - MA6 and barbed end (BE) labelled;
- b) Views of mother filament MA1 - MA6 subunit model depicted as simulated 5 Å resolution EM density;
- c) Views of difference density calculated by subtracting actin model density b) from cryo-EM reconstruction density a). Note that density persists for the Arp2/3 complex, daughter filament and the mother filament reconstruction boundaries because these were not included in the simulated model;
- d) Views of simulated cortactin binding (green model) on mother filament based on observed daughter filament interaction and superimposed on the difference density. The absence of density in the calculated difference density corresponding to cortactin supports the conclusion that there is no (or little) cortactin bound to the mother filament in our reconstruction.

We therefore conclude that the most geometrically probable and most parsimonious explanation is that once the new daughter filament has been nucleated, all the central repeats bind along the same daughter filament, as we have modelled. We are concerned that inclusion of data and discussion in relation to this point in the manuscript would be confusing for readers but would be happy to follow editorial guidance on this.

Additionally, this structure is an adaptation of two recent papers that reconstructed cryo-EM structures of Arp2/3 on an actin branch (Chou et al., 2022; Ding et al., 2022). This makes that the binding of CTTN to the branch is the main novelty of the presented structure. Because of this, it is a pity that the authors were not able to resolve binding of additional CTTN repeat domains, which would have increased the impact of the structure.

5) Our visualization of cortactin at an Arp2/3 branch for the first time is indeed the novelty of our current study, as also endorsed by the other reviewers. This has provided unique insight into its mechanism of daughter filament stabilization at branches and synergy with Class 1 NPFs. Optimizing the cryo-EM sample for predominantly very short daughter filaments was very important for the isotropic distribution of those daughter filaments in the ice and the resulting clear visualization of cortactin density in our reconstruction. The side effect of this was that the daughter filaments were presumably not sufficiently long to accommodate all the actin-binding repeats of cortactin. We completely agree with the reviewer that visualization of more cortactin will be an important goal for the future. Nevertheless, our structure has allowed us to propose a model for how the repeats bind to F-actin.

1. Based on the methods and Fig 1, it is unclear whether the authors attempted to obtain a structure with full length CTTN and that only the Nta and first central repeat domain could be resolved or that only the N-terminal fragment was used to begin with. The authors should state clearly in the manuscript, Methods and Fig 1 whether the structure they obtained was reconstructed with CTTN 1-177 only or with full length CTTN.

6) We apologize for not being clear and have updated the text to specify this point in the main text (lines 127-128), the legend for Fig. 1, Methods and in the newly added flow chart in Extended Data Fig. 1c (in response to point 11).

In case of the latter, they should mention that they were only able to resolve part of the full length protein. Also, is this partial CTTN structure the result of the length of the branches that were obtained because of the high capping protein concentration?

7) The text we have added to the main text mentioned in point 6 (lines 126-128), also addresses this question. Indeed, following the strategies of other studies, and as described on lines 78-79, we added capping protein to limit daughter filament length to enable the isotropic distribution of those daughter filaments in the cryo-EM sample, which is required for higher resolution structure determination.

2. To point 1, the authors deviated from the method established in Chou et al. (2022) to assemble branches and included SPIN90. Can the SPIN90 bound particles be identified and if so, can more CTTN central repeats be resolved? Why was SPIN90 included?

8) We compared and adapted the reconstitution methods from several recently published paper, in particular Ding et al (2022) PMID 35622886, which incorporated SPIN90 to facilitate the generation of mother filaments for branch formation, while using a relatively low concentration of actin to limit *de novo* filament nucleation. Our previous Methods text referred specifically to methods using *S. pombe* Arp2/3 complex, which we have now corrected to include other relevant references with apologies for the confusion (p10). In particular, the method of Ding et al worked well in our hands, but we subsequently focused optimization of our cryo-EM sample on short branch formation, where SPIN90 is not present. 2D analysis of our dataset did not provide evidence of a distinct class average representing SPIN90-Arp2/3 at filaments ends (former Extended Data Fig. 1d, now Extended Data Fig. 1e), showing that these complexes are sufficiently scarce that a structure could not be determined. We agree that this could be an interesting approach to further explore cortactin structure in the future.

3. Fig 3: The authors should also include the actin residues that constitute the other side of the hydrophilic and hydrophobic contacts at the CTTN-F-actin interface.

9) Thank you for this suggestion – we have added these annotations on the left side of Fig. 3d and updated the Figure legend accordingly.

4. Fig 3d: Please indicate the lysines that have been reported to undergo acetylation and show how they interact with F-actin.

10) Thank you for this suggestion. Lysines shown to undergo acetylation are now indicated by * in the sequence alignment in Fig. 3d, with the figure legend updated accordingly. We have modelled the actin interaction with repeat 1 in our structure to highlight the likely inhibitory consequence of acetylation for actin binding in a new Extended Data Fig. 6, and have included some additional text on p5 to expand on this important point.

Minor concerns:

1. The authors should include a flow chart in Fig 1, showing how the EM samples were prepared, including all the other proteins that were used to preassemble branches.

11) Thank you for the suggestion. Such a flow chart is now newly included as Extended Data Fig. 1c, with a reference to it also added to the relevant Methods text (p10).

2. In Fig 1 and 2, the color of CTTN is referred to as “lime”, but looks more like green. It would be better to just call it “green”.

12) We didn't mention the colour of cortactin in the legend to Fig. 1 but we have replaced “lime” with “green” in the legend to Fig. 2.

3. Fig 3d: The numbers in the cyan-colored repeat should be indicated in white so they are visible.

13) We have implemented this suggestion (below in the originally submitted version of the Figure): in the printed version of the figure, white text is indeed clearer against the cyan background but, in our opinion, it is quite a bit less clear in the electronic version. To make the text clearer in both versions, we have changed the text colour to grey on both the cyan and yellow backgrounds in the revised version of Fig. 3d.

Also, the CTTN repeat 3 color is different from the model shown in 3c. Please use consistent color coding.

14) Thanks for pointing this out – the colouring is now consistent between panels.

4. Following the flow of the manuscript, Fig 4 A-B would fit better into Fig 3 than Fig. 4.

15) We thank the reviewer for this suggestion. We have constructed the manuscript such that Fig. 3 focuses on observations and modelling of F-actin binding by the cortactin central repeats. In contrast, Fig. 4 synthesizes these ideas together with our analysis of cortactin NtA interaction with activated Arp3 to consider overall consequences for branch stability and synergy with Class 1 NPFs. We have added an additional call out to Fig. 4a (line 152) to emphasise this point. Although we discussed alternative strategies for presenting the data, encouraged by the positive overall response of the other reviewers and readers who provided feedback on our manuscript before submission, we favour retention of the current figure organisation.

Reviewer #3:

This manuscript by Liu et al., for the first time reports a high resolution structure of cortactin stabilized Arp2/3 complex mediated branched actin junction. The structure shows cortactin bound to daughter filament, but not to the mother filament. This is contrary to what was believed in the field. However, this unique observation sheds light into how cortactin stabilizes these branched actin junctions and possibly WDS-Arp2/3 nucleated linear actin filaments. Furthermore, the structure also reveals at unprecedented details the molecular interactions between the NtA of cortactin and Arp3 subunit. Additionally, the structure reveals that cortactin preferentially interacts with Arp3 and not with Arp2 and elucidates the competitive binding of cortactin with Type-I NPF. Besides the NtA region of cortactin, in the structure the first actin binding repeat of cortactin was well resolved. The remaining 5.5 repeats were not resolved in the structure possibly due to the short length of the daughter filament. Overall this a well put together structural study which provides new insight into what was recently shown by Cao et al., in EMBO J. 2023.

Here are some minor comments and suggestions about the manuscript:

1. It is unclear as to how the authors claim that there is preferential affinity of cortactin NtA towards "activated" Arp3? Does that mean that Cortactin has lower binding affinity to inactive Arp2/3 complex? If so, then how did the authors come to this conclusion?

16) As described in Fig.2 and the accompanying text, our structure shows cortactin NtA bound to activated Arp3 at the Arp2/3-mediated branch, which adopts a conformation that is very similar to other activated Arp2/3 complex structures. We modelled this same cortactin NtA conformation on the published structure of Arp3 in inactive Arp2/3 and found that due to differences in conformation in Arp3 between its active and inactive states, structural clashes were generated (depicted in Extended Data Fig. 5e), and the NtA binding surface visualized in our structure was disrupted. From this, we inferred that cortactin has a lower binding affinity for the inactive Arp2/3 complex,

Further, once Arp2/3 is activated, daughter filament subunits will also provide additional binding sites for cortactin central repeats, further enhancing its apparent affinity for activated Arp2/3, as illustrated in Fig. 4a. Although cortactin has been shown biochemically to interact with inactive Arp2/3 (e.g. Weaver et al (2002) PMID: 12176354), we think that cortactin's higher affinity for activated Arp3 also explains its very weak stimulation of Arp2/3, and we have elaborated this point (which reviewer 1 also raised (see response point 1)) in new text in lines 188-191.

2. What could be the reason why the authors failed to observe any cortactin bound to the mother filament, while previous biochemical data showed binding of cortactin to mother actin filaments?

17) Thanks to the reviewer for raising this key point. Previous data – for example in Helgeson et al (2013) PMID:24015358 – have shown that while cortactin can associate with mother filaments in branching assays ($K_d = 5\mu\text{M}$), it binds much more strongly at nascent junctions ($K_d = 0.017\mu\text{M}$). Earlier studies do not have the resolution to directly differentiate between cortactin binding to mother or daughter filaments at these branch junctions. Nevertheless, cortactin has been routinely depicted as binding to mother filaments in literature; this is presumably because the mother filament clearly and intuitively presents F-actin binding sites at nascent

branches whereas these obviously don't exist until after the daughter filament begins to grow. However, only structural studies such as ours could provide sufficiently precise molecular visualization to discriminate between mother and daughter filament binding. Our structure now provides an explanation for the higher affinity for the nascent junction via its interaction with activated Arp3, which also directs the actin binding repeats along the daughter filament. We have now added some additional text (p7) relating to this point.

3. It would be better if the authors also provide some explanation about synergy between cortactin and other NPFs, while also competing with them.

18) Thanks to the reviewer for raising this lack of clarity, for which we apologize. We have edited the text in the relevant paragraph (from line 163) to more clearly explain our findings, both in the context of prior knowledge about cortactin and VCA-containing NPF synergy (also updated in response to reviewer point 1) and functional insights from cell biology.

Overall it is a nice structural study that is of relevance to the field of branched actin nucleation and stability and must be published

Decision Letter, first revision:

Message: Our ref: NSMB-A48008A

2nd Nov 2023

Dear Dr. Moores,

Thank you for submitting your revised manuscript "Cortactin stabilizes actin branches by bridging activated Arp2/3 to its nucleated actin filament" (NSMB-A48008A). It has now been seen by the original referees and their comments are below. The reviewers find that the paper has improved in revision, and therefore we'll be happy in principle to publish it in Nature Structural & Molecular Biology, pending minor revisions to satisfy the referees' final requests and to comply with our editorial and formatting guidelines.

To facilitate our work at this stage, it is important that we have a copy of the main text as a word file. If you could please send along a word version of this file as soon as possible, we would greatly appreciate it; please make sure to copy the NSMB account (cc'ed above).

Sincerely,
Kat

Katarzyna Ciazynska
(she/her)
Associate Editor
Nature Structural & Molecular Biology
<https://orcid.org/0000-0002-9899-2428>

Reviewer #1 (Remarks to the Author):

All previous issues raised have been addressed to my satisfaction - thanks for carefully considering them!

Reviewer #3 (Remarks to the Author):

This is a nice structural study that clearly describes where and how cortactin binds to Arp2/3 complex and the daughter filament and how that can lead to actin branch-junction stabilization. In the revised manuscript the authors have satisfactorily addressed all our

comments and concerns and that of the other reviewers. We have no further objection for this work to be accepted for publication.

Final Decision Letter:**Message** 18th Dec 2023

:
Dear Dr. Moores,

We are now happy to accept your revised paper "Cortactin stabilizes actin branches by bridging activated Arp2/3 to its nucleated actin filament" for publication as an Article in Nature Structural & Molecular Biology.

As soon as your article is published, you can generate your shareable link by entering the DOI of your article here: http://authors.springernature.com/share. Corresponding authors will also receive an automated email with the shareable link

Your paper will be published online soon after we receive proof corrections and will appear in print in the next available issue. You can find out your date of online publication by contacting the production team shortly after sending your proof corrections.

Please note that *Nature Structural & Molecular Biology* is a Transformative Journal (TJ). Authors may publish their research with us through the traditional subscription access route or make their paper immediately open access through payment of an article-processing charge (APC). Authors will not be required to make a final decision about access to their article until it has been accepted. <https://www.springernature.com/gp/open-research/transformative-journals> Find out more about Transformative Journals

Authors may need to take specific actions to achieve [compliance](https://www.springernature.com/gp/open-research/funding/policy-compliance-faqs) with funder and institutional open access mandates. If your research is supported by a funder that requires immediate open access (e.g. according to [Plan S principles](https://www.springernature.com/gp/open-research/plan-s-compliance)) then you should select the gold OA route, and we will direct you to the compliant route where possible. For authors selecting the subscription publication route, the journal's standard licensing terms will need to be accepted, including

[self-archiving policies](https://www.springernature.com/gp/open-research/policies/journal-policies). Those licensing terms will supersede any other terms that the author or any third party may assert apply to any version of the manuscript.

Sincerely,

Katarzyna Ciazynska, PhD
(she/her)
Associate Editor
Nature Structural & Molecular Biology
<https://orcid.org/0000-0002-9899-2428>